# SOFT-TRANSFORMERS FOR CONTINUAL LEARNING

## ABSTRACT

Inspired by *Well-initialized Lottery Ticket Hypothesis (WLTH)*, which provides suboptimal fine-tuning solutions, we propose a novel fully fine-tuned continual learning (CL) method referred to as Soft-TransFormers (Soft-TF). Soft-TF sequentially learns and selects an optimal soft-network or subnetwork for each task. During sequential training in CL, Soft-TF jointly optimizes the weights of sparse layers to obtain task-adaptive soft (real-valued) networks or subnetworks (binary masks), while keeping the well-pre-trained layer parameters frozen. In inference, the identified task-adaptive network of Soft-TF masks the parameters of the pre-trained network, mapping to an optimal solution for each task and minimizing Catastrophic Forgetting (CF) - the soft-masking preserves the knowledge of the pre-trained network. Extensive experiments on Vision Transformer (ViT) and CLIP demonstrate the effectiveness of Soft-TF, achieving state-of-the-art performance across various CL scenarios, including Class-Incremental Learning (CIL) and Task-Incremental Learning (TIL), supported by convergence theory.

## 1 INTRODUCTION

Continual Learning (CL), also known as Lifelong Learning (Thrun, 1995; Rusu et al., 2016; Zenke et al., 2017; Hassabis et al., 2017), is a learning paradigm where a series of tasks are learned sequentially. The principle objective of continual learning is to replicate human cognition, characterized by the ability to learn new concepts incrementally throughout one's lifespan. An optimal continual learning system could facilitate a positive forward and backward transfer, leveraging the knowledge gained from previous tasks to solve new ones, while also updating its understanding of previous tasks with the new knowledge. However, achieving continual learning is challenging due to the occurrence of *catastrophic forgetting* or *catastrophic interference* (McCloskey & Cohen, 1989), a phenomenon where the performance of the model on previous tasks deteriorates significantly when it learns new tasks. This can make it challenging to retain the knowledge acquired from previous tasks, ultimately leading to a decrease in overall performance. To address the issue of catastrophic forgetting during continual learning, numerous conventional approaches have been proposed on Convolutional Neural Networks (CNNs), which can be broadly classified as follows: (1) **Regularization-based methods** (Kirkpatrick et al., 2017a; Chaudhry et al., 2020; Jung et al., 2020; Titsias et al., 2020; Mirzadeh et al., 2021) aim to keep the learned information of past tasks during continual training aided by sophisticatedly designed regularization terms, (2) **Rehearsal-based methods** (Rebuffi et al., 2017; Riemer et al., 2018; Chaudhry et al., 2019a;b; Saha et al., 2021) utilize a set of real or synthesized data from the previous tasks and revisit them, and (3) **Architecture-based methods** (Mallya et al., 2018; Serrà et al., 2018; Li et al., 2019; Wortsman et al., 2020; Kang et al., 2022; 2023) propose to minimize the inter-task interference via newly designed architectural components.

Developing neural network models that leverage large-scaled pre-trained models. i.e., Vision Transformer (ViT) (Dosovitskiy et al., 2020) and Contrastive Language-Image Pre-training (CLIP) (Radford et al., 2021) leads to a new paradigm shift referred to as (4) **Prompt-based methods** in Continual Learning (CL). Prompt-based methods learn continual representations to provide fixed pre-trained transformers with additional instruction. Notably, while L2P (Wang et al., 2022c) stands out as the seminal work that bridges the gap between prompting and continual learning, DualPrompt (Wang et al., 2022b) introduces an innovative approach to affixing complementary prompts to the pre-trained backbone, thereby enabling the acquisition of both task-invariant and task-specific instructions. Additionally, other notable contributions in this field encompass DyTox (Douillard et al., 2022), S-Prompt (Wang et al., 2022a), CODA-P (Smith et al., 2023b), ConStruct-VL (Smith et al., 2023a),

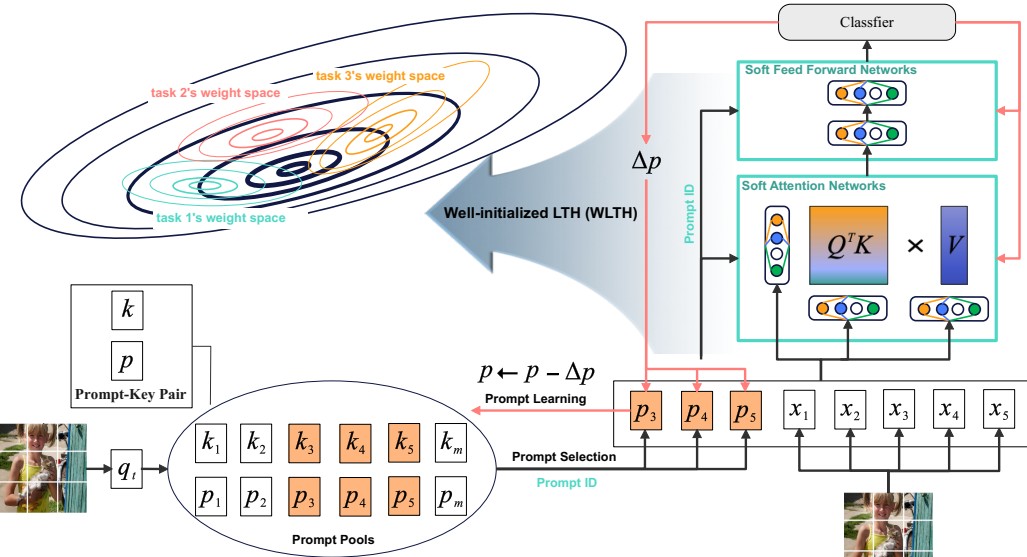

Figure 1: **Soft-TransFormers (Soft-TF)**: the objective is to design a fully fine-tuned model that works well across multiple continual learning settings with incurring task-wise soft network training of attention and feed forward networks, leveraged by WLTH.

ST-Prompt (Pei et al., 2023), and LGCL (Khan et al., 2023). Recently, Qiao et al. (2024) investigated prompt-projection for better generalized continual learners.

With prior developments of representational research, current prompt-based models can be fine-tuned using trained prompts to improve their performance on sequential tasks, and the fixed pre-trained backbone can consistently provide unforgettable base session knowledge. However, prompt-based models come with several disadvantages and limitations. First, the effectiveness of prompt-based CL heavily relies on the quality and design of the sample or task-relevant prompts. Poorly trained prompts could lead to suboptimal performance or tend to be biased. Second, managing and maintaining a large set of prompts can become cumbersome and unmanageable as the number of tasks increases. Lastly, prompt tuning is not as flexible as full fine-tuning. The only prompt-tuning of the pre-trained model cannot capture all the nuances of uncorrelated sequential tasks even though leveraging the well-initialized model pre-trained on large-scale datasets since the well-initialized model provides global solutions rather than task-specific solution. These disadvantages help make informed decisions about when and how to use prompt-based models and explore alternative methods like full fine-tuning or hybrid approaches for more robust and flexible prompt-based continual learning performance.

To overcome the limitations of conventional prompt-based methods, the central focus of this work is to pinpoint the most optimal winning ticket or fine-tuning representations of frozen pre-trained networks such as Transformers in continual learning scenarios. We focus on two main issues when sequential full fine-tuning the pre-trained foundation models: (1) Catastrophic Forgetting (CF) and (2) parameter-efficient fine-tuning CL model. To deploy a practical model to deal with the two points, we suggest a new paradigm for Continual Learning (CL), named *Well-initialized Lottery Ticket Hypotehsis*:

**Well-initialized Lottery Ticket Hypothesis (WLTH).** *A well-initialized dense neural network contains globally minimal solutions that can retain the prior class knowledge while providing room to learn the new class knowledge through isolated fine-tuning of the networks or subnetworks.*

Leveraged by the WLTH, this work proposes a new Soft-TransFormer (Soft-TF) to address fine-tuning with minimal CF, as shown in Figure 1. We could find task-specific soft-networks or subnetworks based on well-trained frozen transformer parameters that incrementally learn task-adaptive weights associated with each task scenario.

Our contributions can be summarized as follows:

- Inspired by *Well-initialized Lottery Ticket Hypothesis (WLTH)*, we propose a novel continual learning method referred to as Soft-TransFormers (Soft-TF), which learns compact task-specific soft-networks or subnetworks from well pre-trained parameters for each task.

- Extensive experiments demonstrate the Soft-TF leads to better generalized continual models than baselines such as DualPrompts, achieving state-of-the-art performances on various class-incremental learning (CIL) and task-incremental learning (TIL) scenarios, as shown in Figure 2.

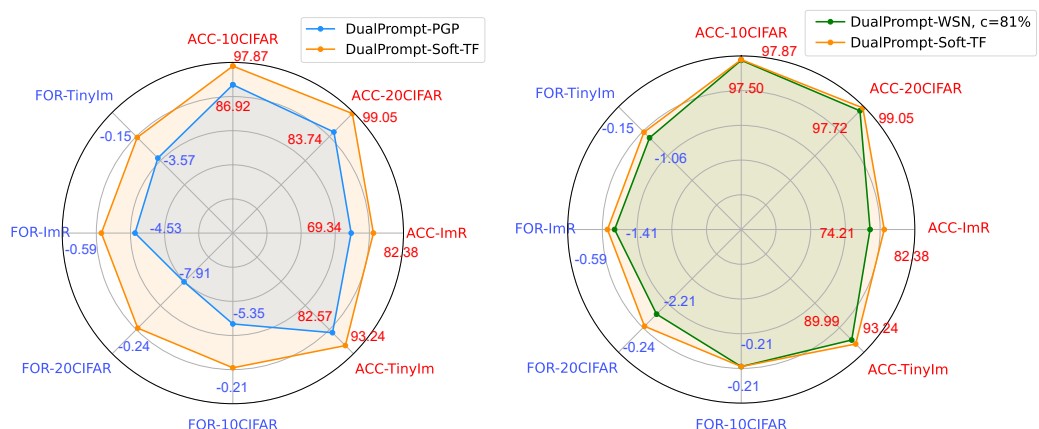

(a) DualPrompt-PGP *v.s.* Soft-TransFormers (Soft-TF) (b) DualPrompt-WSN *v.s.* Soft-TransFormers (Soft-TF)

Figure 2: **Radar Chart of Comparisons** in terms of average accuracy and forgetting between baselines and our SOTA method (Soft-TF). DualPrompt-PGP (Qiao et al., 2024) and DualPrompt-WSN (Kang et al., 2022) (c% sparsity) are baselines for prompt tuning and subnetworks. ACC refers to the average accuracy metric (higher is better). FOR refers to the forgetting metric (lower is better). Different scale standards are adopted for two metrics on benchmark datasets.

## 2 RELATED WORKS

**Continual Learning** (McCloskey & Cohen, 1989; Thrun, 1995; Kumar & Daume III, 2012; Li & Hoiem, 2016) is the challenge of learning a sequence of tasks continuously while utilizing and preserving previously learned knowledge to improve performance on new tasks. Four major approaches have been proposed to tackle the challenges of continual learning, such as catastrophic forgetting. One such approach is *Regularization-based approaches* (Kirkpatrick et al., 2017a; Chaudhry et al., 2020; Jung et al., 2020; Titsias et al., 2020; Mirzadeh et al., 2021), which aim to reduce catastrophic forgetting by imposing regularization constraints that inhibit changes to the weights or nodes associated with past tasks. *Rehearsal-based approaches* (Rebuffi et al., 2017; Chaudhry et al., 2019a;b; Saha et al., 2021; Deng et al., 2021; Sun et al., 2023; Sarfraz et al., 2023; Mai et al., 2021; Lin et al., 2023; Aljundi et al., 2019; Caccia et al., 2021; Chaudhry et al., 2019c; Liang & Li, 2024; Buzzega et al., 2020) store small data summaries to the past tasks and replay them during training to retain the acquired knowledge. Some approaches in this line of work (Shin et al., 2017; Aljundi et al., 2019) accommodate the generative model to construct the pseudo-rehearsals for previous tasks. *Architecture-based approaches* (Mallya et al., 2018; Serrà et al., 2018; Li et al., 2019; Wortsman et al., 2020; Kang et al., 2022; 2023; 2024b;a) use the additional capacity to expand (Xu & Zhu, 2018; Yoon et al., 2018), dynamic representation (Yan et al., 2021; Singh et al., 2020) or isolate (Rusu et al., 2016) model parameters, preserving learned knowledge and preventing forgetting. Rehearsal and architecture-based methods have shown remarkable efficacy in suppressing catastrophic forgetting but require additional capacity for the task-adaptive parameters (Wortsman et al., 2020) or the replay buffers. Recently, *Prompt-based approaches*, an emerging transfer learning technique, harnesses a fixed function of pre-trained Transformer models. This empowers the language model to receive additional instructions for enhancing its performance on downstream tasks. Notably, while L2P (Wang et al., 2022c) stands out as the seminal work that bridges the gap between prompting and continual learning, DualPrompt (Wang et al., 2022b) introduces an innovative approach to affixing complementary prompts to the fixed pre-trained backbone. Here, we introduce a new approach to update the fixed pre-trained parameters through learnable sparse networks under the convergence theory, maximumly enabling the acquisition of task-invariant and task-specific instructions.

**Prompt-based CL.** With recent advances in Vision Transformers (Khan et al., 2022) and prompt-based fine-tuning in NLP (Li & Liang (2021)), Wang et al. (2022c) have shown that interacting

with an ImageNet pre-trained model via prompt learning is a promising approach, L2P (Wang et al., 2022c) for continual learning (DualPrompt (Wang et al., 2022b), DyTox (Douillard et al., 2022), S-Prompt Wang et al. (2022a), CODA-P (Smith et al., 2023b), ConStruct-VL Smith et al. (2023a), ST-Prompt (Pei et al., 2023), and LGCL (Khan et al., 2023)). Recently, Prompt Gradient Projection (PGP) (Qiao et al., 2024), a small set of learnable orthogonal parameters, is appended to the input and enables quick adaptation of a frozen ImageNet pre-trained model to new streaming tasks. Their analysis shows that directly leveraging the pre-trained vision-language model without introducing any learnable parameters is a simple yet promising approach to continual learning. The PGP adopted a joint vision-language model like CLIP (Radford et al., 2021) for continual learning, which presents multiple advantages. It enables catering for practical scenarios with no well-defined task identities and boundaries, and the model is required to adapt to streaming data dynamically in a task-agnostic manner. However, prompt-based models come with several disadvantages. Poorly trained prompts could lead to suboptimal performance or tend to be biased. Moreover, prompt tuning could not capture all nuances of uncorrelated sequential tasks. These disadvantages lead to exploring alternative methods like full fine-tuning or hybrid approaches for more robust and flexible prompt-based model performance. In this work, to alleviate these issues, we investigate a fullly fine-tuning of well-pre-trained transformers on training soft networks and finding competitive subnetworks.

## 3 PREREQUISITES

We start with conventional prompt-based continual learning methods using Vision Transformer (ViT) (Dosovitskiy et al., 2020) and Contrastive Language-Image Pre-training (CLIP) (Radford et al., 2021) in Class Incremental Learning (CIL) and Task Incremental Learning (TIL) scenarios.

### 3.1 PRELIMINARIES

**Problem Statement**. Continual Learning (CL) involves training deep neural networks (DNN) on time-variant data represented as a sequence of tasks, $\mathcal{D} = \{\mathcal{D}_1, \cdots, \mathcal{D}_{\mathcal{T}}\}$. Each $t$-th task, $\mathcal{D}_t = \{(\boldsymbol{x}_i^t, y_i^t)_{i=1}^{n_t}\}$ consists of $n_t$ tuples where $\boldsymbol{x}_i^t \in \mathcal{X}_t$ is an input sample and $y_i^t \in \mathcal{Y}_t$ is the corresponding label. When a task $\mathcal{X}_t$ arrives, a model $f_\theta$ is trained for the current task, while data from previous tasks is inaccessible. This work focuses primarily on class incremental learning (CIL), in which the task-ID is not given during inference.

**Soft & Subnetworks** have been explored in continual learning through two notable approaches. One approach, known as supermasks (Wortsman et al., 2020), produces outputs by $\boldsymbol{p} = f(\boldsymbol{x}, \boldsymbol{w} \odot \boldsymbol{m})$, where $\odot$ denotes elementwise multiplication. In this method, the weights $\boldsymbol{w}$ remain fixed at their initialization, with bias terms set to $\boldsymbol{0}$ and other parameters initialized to $\pm c$ with equal probability where the constant $c$ is the standard deviation of the corresponding Kaiming normal distribution (He et al., 2015). Another line of work includes WSN (Kang et al., 2022) and SoftNet (Kang et al., 2023), which jointly learn the model weights $\boldsymbol{w}$ and task-adaptive subnetworks $\boldsymbol{m}$. The parameter-efficient reusable subnetworks are obtained by iteratively selecting the top-$c\%$ of the weights based on an importance score $\boldsymbol{s}$ at each layer. WSN has primarily demonstrated its effectiveness in Convolutional Neural Networks (CNNs). However, its pruning mechanism for pre-trained Transformers, such as ViT, remains unexplored. To discover the competitive sparseness in Transformers, we detail the WSN-style task-adaptive fine-tuning and the learnable soft-networks $\boldsymbol{m}$ of Transformers, presenting these adaptations for the first time with empirical observations. The soft-networks originate from learned parameters distributed with $\mu \approx 1.0$ & various variances, as stated in Figure 6.

### 3.2 PROMPT-BASED CLASS INCREMENTAL LEARNING (CIL)

A simple yet effective prompt-based (prompt-tuning) CIL model: Learning to Prompt (L2P) (Wang et al., 2022c) is first proposed. In this model, a prompt $p$, a tiny set of trainable tokens combined with image features, is fed into the Vision Transformer (ViT) to help the model resist forgetting. To select suitable prompts for task-specific training, L2P utilizes a prompt pool $P$ containing numerous prompt-key pairs, $\{p_t, k_t\}_{t=1}^{\mathcal{T}}$, where $p_t \in \mathbb{R}^{1 \times D}$ represents the $t$-th task prompt, $k_t$ represents the $t$-th coresponding task key, and $\mathcal{T}$ is the total number of prompt-key pairs.

Building on L2P, DualPrompt (Wang et al., 2022b) divided the prompts into expert (E-) prompts and general (G-) prompts for distinct features learning. DualPrompt also replaced prompt-tuning

with prefix-tuning, which was successfully proven in NLP. DyTox (Douillard et al., 2022) designed a novel task attention block that utilized task tokens to infer task identifiers. Coda-Prompt (Smith et al., 2023b) replaced the prompt pool with a decomposed prompt, represented by a weighted sum of learnable prompt components, which optimized itself in an end-to-end fashion, providing high plasticity. LGCL (Khan et al., 2023) introduced text information into the learning of prompt pool, improving performance without any additional learnable parameters.

Recently, Qiao et al. (2024) introduced Prompt Gradient Projection (PGP), which applies an orthogonal condition on the prompt gradient to reduce forgetting via the self-attention mechanism in ViT effectively. Although various prompt-based continual learners have demonstrated state-of-the-art performance, they do not explicitly model task-specific fine-tuning and forgetting within the continual learning framework. In this work, we address task-specific fine-tuning and gradient-based task identification in CIL and TIL scenarios by leveraging prompt-tuning and learnable sparse networks.

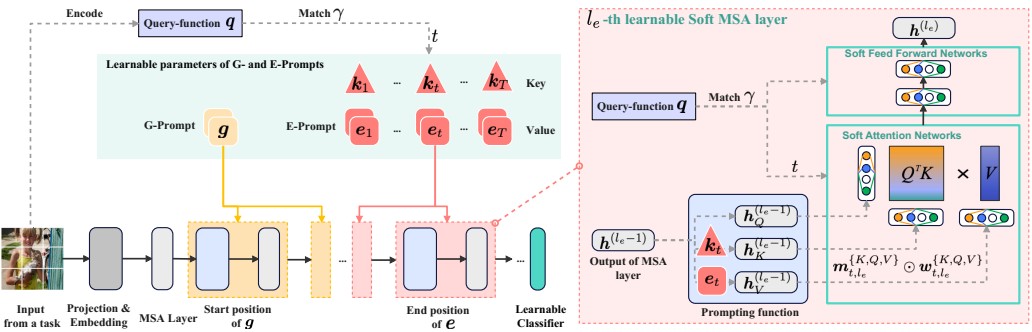

Figure 3: **Soft-TransFormers (Soft-TF)**: At training time, the E-Prompt and the Soft-network are selected according to task identity, and the selected G-Prompt, E-Prompt, and the Soft-networks (Soft-Attention and Feed Forwards) are trained together with a classifier. At test time, an input is transformed by a query function (Prompt ID) or task identifier (Gradient ID) to match the closest task key $\boldsymbol{k}_t$, E-prompt $\boldsymbol{e}_t$ and Soft-networks $\boldsymbol{m}_t^{\{K,Q,V\}}$. Note task identifier is depicted in Section 5.

## 4 TRANSFORMER WITH LEARNABLE SUBNETWORKS

In this section, we explain how Soft-TransFormers (Soft-TF) leverage learnable soft-networks to train sequential tasks while keeping the well-pretrained model parameters fixed. To introduce our novel Soft-TF and provide a clearer understanding, we draw on a partial explanation of DualPrompt.

### 4.1 SOFT-MSA LAYERS

To address the task-specific fine-tuning of the pre-trained model, such as ViT, this work proposes a new Soft-TransFormer (Soft-TF), as illustrated in Figure 1. The proposed Soft-TF consists of a conventional neural network, like a multilayer transformer with multihead attention and forward networks. Using well-trained transformer parameters, we could discover task-specific soft-networks, as depicted in Figure 3. The Soft-TF incrementally learns model weights and task-adaptive soft-masks with well-pre-trained and soft-network parameters $\boldsymbol{m}$.

Given a pre-trained parameter $\boldsymbol{\theta}$ and learnable soft-parameters $\boldsymbol{m}$, Soft-ViT is represented as $f_{\boldsymbol{\theta} \odot \boldsymbol{m}}$, consisting of $N$ consecutive soft-MSA layers. We extend the notation by denoting the input embedding feature of the $l_*$-th learnable soft-MSA layer as $\boldsymbol{h}^{(l_*)}$, where $l_* = 1, 2, \ldots, N$, and $l_*$ can refer to either the G-Prompt layer $l_g$ or the E-Prompt layer $l_e$. Note that while the pre-trained parameters $\boldsymbol{\theta}$ remain fixed, the soft-parameters $\boldsymbol{m}$ are updated to provide task-specific solutions.

**G-prompt.** $\boldsymbol{g} \in \mathbb{R}^{L_g \times D}$ with sequence length $L_g$ and embedding dimension $D$, is a shared parameter for all tasks. G-Prompt is attached to the $l_g$-th MSA layer to transform $\boldsymbol{h}^{(l_g)}$ via a prompting function as follows:

$$\boldsymbol{h}_g^{(l_g)} = f_{\boldsymbol{\theta}}^{prompt}(\boldsymbol{g}, \boldsymbol{h}^{(l_g)}), \tag{1}$$

where $f_{\boldsymbol{\theta}}^{prompt}$ defines the approach for attaching the prompt to the hidden embeddings.

**E-prompt & Soft-networks.** $\boldsymbol{e} = \{\boldsymbol{e}_t\}_{t=1}^{\mathcal{T}}$ is a set of task-dependent parameters, where $\boldsymbol{e}_t \in \mathbb{R}^{L_e \times D}$ has as sequence length of $L_e$ and the same embedding dimension $D$ as the G-prompt, and $\mathcal{T}$ is the total number of tasks. Unlike the shared G-prompt, each $\boldsymbol{e}_t$ is associated with a task-specific key $\boldsymbol{k}_t \in \mathbb{R}^D$, which is also a learnable parameter aimed at capturing representative features of a task. For an input example from the $t$-th task, to attach E-prompt to the $l_e$-th soft-MSA layer, we apply the prompting function in a similar way:

$$\boldsymbol{h}_e^{(l_e)} = f_{\boldsymbol{\theta} \odot \boldsymbol{m}}^{prompt}(\boldsymbol{e}_t, \boldsymbol{h}^{(l_e)}). \tag{2}$$

## 4.2 Prompts with Learnable Subnetworks

G- and E-prompts, along with learnable soft-networks, encode specific types of instructions during training with the backbone and work together to guide the model's predictions during inference. We have demonstrated the method for attaching prompts and learnable soft-networks to a single soft-MSA layer. Similarly to the approach taken in DualPrompt (Wang et al., 2022b), we also investigate layers of E-prompts with learnable soft-networks $\boldsymbol{m}$, while utilizing the layers designated for G-prompts.

**Layers of G- and E-Prompts**. We use the multilayered extension of both types of prompts: $\boldsymbol{g} = \{\boldsymbol{g}^{(l_g)}\}_{l_g=start_g}^{end_g}$, where $\boldsymbol{g}^{(l_g)} \in \mathbb{R}^{L_g \times D}$ represents the G-prompt attached to the $l_g$-th MSA layer. Similarly, we define $\boldsymbol{e}_t = \{\boldsymbol{e}_t^{(l_e)}\}_{l_e=start_e}^{end_e}$ for the $l_e$-th conventional MSA layer. In this configuration, the G-prompt $\boldsymbol{g}^{(l_g)}$ is attached from the $start_g$-th to the $end_g$-th conventional MSA layers, and the E-prompt $\boldsymbol{e}_t^{(l_e)}$ is attached to the $[start_e, end_e]$-th soft-MSA layers, ensuring that there is no overlap between them. In our experiments, we follow the $l_g \notin [start_e, end_e]$ ($l_g = [1, 2]$) settings used in DualPrompt and empirically search for the optimal $[start_e, end_e]$ layers for the learnable subnetworks through ablation studies.

**Learnable Soft-networks**. The prompting function $f_{\boldsymbol{\theta} \odot \boldsymbol{m}}^{prompt}$ determines how prompts ($\boldsymbol{p}$) are combined with fine-tuned soft ($\boldsymbol{\theta} \odot \boldsymbol{m}$) embedding features. From another perspective, $f_{\boldsymbol{\theta} \odot \boldsymbol{m}}^{prompt}$ directly influences the interaction between high-level instructions in the prompts and low-level representations. Therefore, we believe that a well-designed prompting function, along with task-specific parameters, is crucial for optimizing overall continual learning performance.

Specifically, applying a prompting and fine-tuning function $f_{\boldsymbol{\theta} \odot \boldsymbol{m}}^{prompt}$ can be seen as modifying the inputs to the soft-MSA layers. Let the input to the soft-MSA layer be $\boldsymbol{h} \in \mathbb{R}^{L \times D}$, and denote the input query, key, and values for the soft-MSA layer as $\boldsymbol{h}_Q$, $\boldsymbol{h}_K$, and $\boldsymbol{h}_V$, respectively. A soft-MSA layer is defined by the following equation:

$$\text{MSA}(\boldsymbol{h}_Q, \boldsymbol{h}_K, \boldsymbol{h}_V) = \text{Concat}(\boldsymbol{h}_1, \cdots, \boldsymbol{h}_i, \cdots, \boldsymbol{h}_n)\boldsymbol{w}^O \odot \boldsymbol{m}^O$$
$$\text{where } \boldsymbol{h}_i = \text{Attention}(\boldsymbol{h}_Q(\boldsymbol{w}_i^Q \odot \boldsymbol{m}^Q), \boldsymbol{h}_K(\boldsymbol{w}_i^K \odot \boldsymbol{m}^K), \boldsymbol{h}_V(\boldsymbol{w}_i^V \odot \boldsymbol{m}^V)), \tag{3}$$

where $\boldsymbol{w}_i^O$, $\boldsymbol{w}_i^Q$, $\boldsymbol{w}_i^K$, and $\boldsymbol{w}_i^V$ are fixed projection matrices while $\boldsymbol{m}^O$, $\boldsymbol{m}^Q$, $\boldsymbol{m}^K$, and $\boldsymbol{m}^V$ are learnable parameters. $s$ is the number of heads. In ViT, $\boldsymbol{h}_Q = \boldsymbol{h}_K = \boldsymbol{h}_V$. Here, we define a unified prompt parameter with a sequence length of $L_p$, such as $\boldsymbol{p} \in \mathbb{R}^{L_p \times D}$ for a single-layered G- or E-prompt.

## 4.3 Fine-tuning on Well-initialized Parameters

In this framework, we concatenate the prompts $\boldsymbol{p}_t$ and the embedding sequence $\boldsymbol{x}_t$, i.e., inputs from $t$-th task, along the embedding dimension: $\boldsymbol{z}_t = [\boldsymbol{p}_t; \boldsymbol{x}_t]$. With the weights of $\boldsymbol{w}^Q \odot \boldsymbol{m}^Q, \boldsymbol{w}^K \odot \boldsymbol{m}^K, \boldsymbol{w}^V \odot \boldsymbol{m}^V$, the soft-transformer takes query ($\boldsymbol{q}_t = (\boldsymbol{w}^Q \odot \boldsymbol{m}^Q)\boldsymbol{z}_t$) and key ($\boldsymbol{k}_t = (\boldsymbol{w}^K \odot \boldsymbol{m}^K)\boldsymbol{z}_t$) as input of the soft-MSA layer. The soft-attention matrix is then given by:

$$\boldsymbol{a}_t = \text{softmax}\left(\frac{\boldsymbol{q}_t \boldsymbol{k}_t^T}{\sqrt{D/n}}\right) \tag{4}$$

where we focus on $\boldsymbol{q}_t \boldsymbol{k}_t^T = (\boldsymbol{w}^Q \odot \boldsymbol{m}^Q)\boldsymbol{z}_t$ and $\boldsymbol{z}_t^T(\boldsymbol{w}^K \odot \boldsymbol{m}^K)^T$. First, the trainable prompt parameters can be denoted as:

$$\boldsymbol{z}_t \cdot \boldsymbol{z}_t^T = \begin{bmatrix} \boldsymbol{p}_t \\ \boldsymbol{x}_t \end{bmatrix} [\boldsymbol{p}_t \ \boldsymbol{x}_t] = \begin{bmatrix} \boldsymbol{p}_t \boldsymbol{p}_t^T & \boldsymbol{p}_t \boldsymbol{x}_t^T \\ \boldsymbol{x}_t \boldsymbol{p}_t^T & \boldsymbol{x}_t \boldsymbol{x}_t^t \end{bmatrix} \tag{5}$$

Second, the trainable soft-attention layer's parameters with $\boldsymbol{m}^Q$ and $\boldsymbol{m}^K$ are as follows:

$$\begin{cases} \boldsymbol{m}^Q \cdot \boldsymbol{p}_t \boldsymbol{p}_t^T \cdot (\boldsymbol{m}^K)^T, \\ \boldsymbol{m}^Q \cdot \boldsymbol{x}_t \boldsymbol{p}_t^T \cdot (\boldsymbol{m}^K)^T, \\ \boldsymbol{m}^Q \cdot \boldsymbol{p}_t \boldsymbol{x}_t^T \cdot (\boldsymbol{m}^K)^T, \\ \boldsymbol{m}^Q \cdot \boldsymbol{x}_t \boldsymbol{x}_t^T \cdot (\boldsymbol{m}^K)^T. \end{cases} \qquad (6)$$

where $\boldsymbol{w}^Q$ and $\boldsymbol{w}^K$ are frozen and unchanged during training and test.

### 4.4 THE OPTIMIZATION OF SOFT-TRANSFORMERS

The overall process of the Soft-TransFormers (Soft-TF) during training and testing is described as Algorithm 1 and Algorithm 2. We denote the architecture with attached prompts as $f_{\boldsymbol{g},\boldsymbol{e}_t,\boldsymbol{m}_t}$. The input $\boldsymbol{x}$ of the $t$-th task is transformed using $f_{\boldsymbol{g},\boldsymbol{e}_t,\boldsymbol{m}_t}$ and then passed to the classification head $f\phi$, parameterized by $\phi$, for prediction. Finally, we train the two prompts, the task keys, the soft-attention parameters, and the newly-initialized classification head in an end-to-end manner:

$$\min_{\boldsymbol{g},\boldsymbol{e}_t,\boldsymbol{m}_t,\boldsymbol{k}_t,\phi} \mathcal{L}(f_\phi(f_{\boldsymbol{g},\boldsymbol{e}_t,\boldsymbol{m}_t}(\boldsymbol{x})),y) + \lambda \mathcal{L}_{\text{match}}(\boldsymbol{x},\boldsymbol{k}_t), \quad \boldsymbol{x} \in \mathcal{D}_t, \qquad (7)$$

Here, $\mathcal{L}$ represents the cross-entropy loss, and $\mathcal{L}_{\text{match}}(\boldsymbol{x},\boldsymbol{k}_t) = \gamma(\boldsymbol{q}(\boldsymbol{x}),\boldsymbol{k}_t)$ denotes the matching loss, where $\boldsymbol{q}(\boldsymbol{x}) = f(\boldsymbol{x})[0]$ corresponds to the feature vector associated with the **[class]** token (Dosovitskiy et al., 2020; Wang et al., 2022b), and $\gamma$ is the cosine similarity. The scalar $\lambda$ serves as a balancing factor between the losses; here, we follow the same DualPrompt setting as a baseline.

**Analysis of Soft-TF for Convex-Lipschitz Functions**. To analyze the convergence rate of the Soft-Transformer, we focus on the case of convex-Lipschitz functions. Let $\boldsymbol{w}^* = \{\boldsymbol{g}^*, \boldsymbol{e}_t^*, \boldsymbol{m}_t^*\}$ be any vector, and let $B$ be an upper bound on $||\boldsymbol{w}^*||$ when $\boldsymbol{w}^{(1)} = \boldsymbol{0}$, or $\boldsymbol{w}^{(1)}$ is an initial state. It is helpful to consider $\boldsymbol{w}^*$ as the minimizer of $f(\boldsymbol{w})$, although the following analysis applies to any $\boldsymbol{w}^*$.

We derive an upper bound on the sub-optimality of our solution relative to $\boldsymbol{w}^*$, specifically $f(\bar{\boldsymbol{w}}) - f(\boldsymbol{w}^*)$, where $\bar{\boldsymbol{w}} = \frac{1}{T}\sum_{t=1}^{T} \boldsymbol{w}^{(t)}$. By the definition of $\bar{\boldsymbol{w}}$ and applying Jensen's inequality, we obtain the following (see Appendix A.1 stated in detail):

$$f(\bar{\boldsymbol{w}}) - f(\boldsymbol{w}^*) = f\left(\frac{1}{T}\sum_{t=1}^{T} \boldsymbol{w}^{(t)}\right) - f(\boldsymbol{w}^*)$$

$$\leq \frac{1}{T}\sum_{t=1}^{T}\left(f(\boldsymbol{w}^{(t)})\right) - f(\boldsymbol{w}^*) \qquad (8)$$

$$= \frac{1}{T}\sum_{t=1}^{T}\left(f(\boldsymbol{w}^{(t)}) - f(\boldsymbol{w}^*)\right).$$

For every $t$, because of the convexity of $f$, we have that

$$f(\boldsymbol{w}^{(t)}) - f(\boldsymbol{w}^*) \leq \left\langle \boldsymbol{w}^{(t)} - \boldsymbol{w}^*, \nabla f(\boldsymbol{w}^{(t)}) \right\rangle \qquad (9)$$

Combining the preceding, we obtain

$$f(\boldsymbol{w}^{(t)}) - f(\boldsymbol{w}^*) \leq \frac{1}{T}\sum_{t=1}^{T} \left\langle \boldsymbol{w}^{(t)} - \boldsymbol{w}^*, \nabla f(\boldsymbol{w}^{(t)}) \right\rangle \qquad (10)$$

To bound the right-hand side of the above formula, we rely on the following lemma:

**Lemma 4.1.** *Let $\boldsymbol{v}_1, \cdots, \boldsymbol{v}_t, \cdots, \boldsymbol{v}_T$ be an arbitrary sequence of vectors, such as the $t$-th task gradients $\boldsymbol{v}_t = \nabla f(\boldsymbol{w})^t$. Consider any algorithm with a well-initialized (well pre-trained Transformer from WLTH) starting point $\boldsymbol{w}^{(1)} \neq \boldsymbol{0}$ and an update rule of the form:*

$$\boldsymbol{w}^{(t+1)} = \boldsymbol{w}^{(t)} - \eta \boldsymbol{v}_t \qquad (11)$$

*satisfies with* $||\boldsymbol{w}^{(1)} - \boldsymbol{w}^*||^2 = ||\boldsymbol{w}^{(T+1)} - \boldsymbol{w}^*||^2$

$$\sum_{t=1}^{T} \left\langle \boldsymbol{w}^{(t)} - \boldsymbol{w}^*, \boldsymbol{v}_t \right\rangle \leq \frac{1}{2\eta_m} ||\boldsymbol{w}_m^{(T+1)} - \boldsymbol{w}^*||^2 + \frac{\eta_m}{2} \sum_{t=1}^{T} ||\boldsymbol{v}_t||^2$$

$$< \frac{1}{2\eta_p} ||\boldsymbol{w}_p^{(T+1)} - \boldsymbol{w}^*||^2 + \frac{\eta_p}{2} \sum_{t=1}^{T} ||\boldsymbol{v}_t||^2 \qquad (12)$$

*where $\boldsymbol{w}_m \neq \boldsymbol{w}_p$ since $\boldsymbol{m}$ is learnable parameters in Soft-TransFormers. Specifically, we could assume that $\boldsymbol{w}_m = (\boldsymbol{w}^Q \odot \boldsymbol{m}^Q) \cdot \boldsymbol{x}\boldsymbol{p}^T \cdot (\boldsymbol{w}^K \odot \boldsymbol{m}^K)^T$ and $\boldsymbol{w}_p = (\boldsymbol{w}^Q \odot \boldsymbol{1}^Q) \cdot \boldsymbol{x}\boldsymbol{p}^T \cdot (\boldsymbol{w}^K \odot \boldsymbol{1}^K)^T$ of [Equation 6](#) and $\boldsymbol{w}^Q, \boldsymbol{w}^K$ are here frozen pre-trained parameters. For every $B_m < B_p < B, \rho > 0$ where $B_m = ||\boldsymbol{w}_m^{(T+1)} - \boldsymbol{w}^*||$ and $B_p = ||\boldsymbol{w}_p^{(T+1)} - \boldsymbol{w}^*||$, if for all t we have that $||\boldsymbol{v}_t \leq \rho||$ and we set $\eta \approx \eta_m \approx \eta_p = \sqrt{\frac{B^2}{\rho^2 T}}$ with large enough T, then for every $\boldsymbol{w}^*$ with $||\boldsymbol{w}^{(T+1)} - \boldsymbol{w}^*|| \leq B$ we have*

$$\frac{1}{T} \sum_{t=1}^{T} \left\langle \boldsymbol{w}^{(t)} - \boldsymbol{w}^*, \boldsymbol{v}_t \right\rangle \leq \frac{B_m \rho}{\sqrt{T}} < \frac{B_p \rho}{\sqrt{T}} < \frac{B\rho}{\sqrt{T}}. \qquad (13)$$

## 5 EXPERIMENTS

We validate our method on several benchmark datasets against continuous learning baselines in Class-Incremental Learning (CIL) and Task-Incremental Learning (TIL).

### 5.1 EXPERIMENTAL SETTINGS

**Datasets.** We evaluate our method mainly on 1) 10/20-Split-CIFAR100 (Krizhevsky et al., 2009), constructed by splitting the 100 classes into 10 tasks/20 tasks. 2) 10-Split-TinyImageNet (Abai & Rajmalwar, 2019), constructed by splitting the 200 classes into 10 tasks. 3) 10-Split-ImageNet-R (Hendrycks et al., 2021), constructed by splitting the 200 classes into 10 tasks. To show our effectiveness, we additionally compare our method with the baselines on 5-Split-CUB200 and 10-Split-TinyImageNet. The detailed experimental settings are depicted in the Supplementary.

**Implementation.** For fair comparisons, we set L2P (Wang et al., 2022c), DualPrompt (Wang et al., 2022b), CLIP (Radford et al., 2021), and PGP (Qiao et al., 2024) as our baselines. We follow experimental settings Qiao et al. (2024) entirely.

**Baselines.** To validate the powerfulness of our method, we compare our results with various CIL baselines including ICaRL (Rebuffi et al., 2017), BiC (Wu et al., 2019), DER++ (Buzzega et al., 2020), LWF (Li & Hoiem, 2017), EWC (Kirkpatrick et al., 2017b), DER+MCG (Cai et al., 2023), and DualPrompt-PGP (Qiao et al., 2024). In addition, we investigate subnetwork solutions such as WSN (Kang et al., 2022) (obtained by selecting top-$c\%$ of weight scores while fixing pre-trained parameters) and SoftNet (Kang et al., 2023) (acquired by selecting top-$c\%$ of weight scores as major tickets while setting $100.0 - \text{top-}c\%$ as minor tickets) in Vision Transformers (ViT) using prompt tuning methods. We adopt average accuracy (ACC) and forgetting (FOR) as our validation metrics (Wang et al., 2022b; Qiao et al., 2024).

**Task Inference.** At the inference time, we infer task identity for arbitrary pieces of task samples $\boldsymbol{x}$ for finding the proper task nuances and demonstrating full fine-tuning results. We summarize the following two methods:

- **Prompt ID**: For a test example $\boldsymbol{x}$, we simply choose the best matched task index via $\text{argmin}_t \gamma(q(\boldsymbol{x}), \boldsymbol{k}_t)$.
- **Gradient ID**: To infer the task identity, we follow SupSup's one-shot task inference (Wortsman et al., 2020). In short, we assign each learned subnetwork $\boldsymbol{m}_t$ a weight $\alpha_t$ such that $\sum_t \alpha_t = 1$ and $\alpha_t = 1/\mathcal{T} > 0$ when evaluating all seen tasks. Given an example data point of batch $\boldsymbol{x} \in \boldsymbol{b}$ to classify, we can compute the loss as $\mathcal{L} = \mathcal{H}(f_{\boldsymbol{\theta} \odot (\sum_t \alpha_t \boldsymbol{m}_t)}^{prompt}(\boldsymbol{x}))$ where $f_{\boldsymbol{\theta}}^{prompt}(\boldsymbol{x})$ is the pre-trained model which outputs logits and $\mathcal{H}$ is the entropy function. From here our inferred task is simply $\hat{t} = \text{argmin}_t \frac{\partial \mathcal{H}}{\partial \alpha_t}$.

Table 1: **Performances of Class Incremental Learning (CIL)** in terms of accuracy and forgetting on 10/20-Split-CIFAR100 and 10-Split-ImageNet-R. Exemplar means the total buffer size for rehearsal methods.

| Method | Exemplar | Task ID | 10-Split-CIFAR100 | | 20-Split-CIFAR100 | | 10-Split-ImageNet-R | |
|---|---|---|---|---|---|---|---|---|
| | | | ACC(↑) | Forget(↓) | ACC(↑) | Forget(↓) | ACC(↑) | Forget(↓) |
| BiC | 5,000 | - | 81.42 | 17.31 | 73.02 | 6.23 | 64.63 | 22.25 |
| DER++ | 5,000 | - | 83.94 | 14.55 | - | - | 66.73 | 20.67 |
| iCaRL | 5,000 | - | 66.00 | 5.33 | 78.02 | 5.80 | - | - |
| DER+MCG | 2,000 | - | 67.62 | 14.64 | 65.84 | 13.72 | - | - |
| BiC | 1,000 | - | 66.11 | 35.24 | 63.12 | 21.89 | 52.14 | 36.70 |
| DER++ | 1,000 | - | 61.06 | 39.87 | - | - | 55.47 | 34.64 |
| iCaRL | 1,000 | - | 61.25 | 14.19 | 71.32 | 15.98 | - | - |
| FT | - | - | 33.61 | 86.87 | 33.52 | 53.69 | 28.87 | 63.80 |
| EWC | - | - | 47.01 | 33.27 | 36.73 | 35.19 | 35.00 | 56.16 |
| LWF | - | - | 60.69 | 27.77 | 39.12 | 57.91 | 38.54 | 52.37 |
| L2P* | - | Prompt ID | 83.77 | 6.63 | 71.29 | 13.96 | 60.44 | 9.00 |
| L2P-PGP* | - | Prompt ID | 84.34 | 5.59 | 76.12 | **13.26** | 61.40 | 8.03 |
| L2P-PGP-**Soft-TF** | - | Prompt ID | 86.26 | **4.79** | 76.17 | 15.77 | **69.80** | **5.13** |
| L2P-PGP-**Soft-TF** | - | **Gradient ID** | 86.46 | 4.87 | **77.67** | 15.84 | 69.56 | 5.28 |
| DualPrompt-PGP | - | Prompt ID | 86.92 | 5.35 | 83.74 | 7.91 | 69.34 | 4.53 |
| DualPrompt-PGP-**Soft-TF** | - | Prompt ID | 92.41 | 2.44 | 95.14 | 1.90 | 74.65 | 4.39 |
| DualPrompt-PGP-**Soft-TF** | - | **Gradient ID** | **92.92** | **2.34** | **95.89** | **1.64** | **81.45** | **2.89** |
| DualPrompt | - | Prompt ID | 86.50 | 5.77 | 82.98 | 8.20 | 68.13 | 4.46 |
| DualPrompt-**Soft-TF** | - | Prompt ID | 91.77 | 3.37 | 94.43 | 2.02 | 74.70 | 6.46 |
| DualPrompt-**Soft-TF** (SOTA) | - | **Gradient ID** | **97.87** | **0.21** | **99.05** | **0.24** | **82.38** | **0.59** |
| Upper-Bound of Soft-TF | - | - | 93.90 | - | 93.90 | - | 80.21 | - |

Table 2: **Performances of Class Incremental Learning (CIL)** in terms of Pretained-dataset (ImageNet-21K, SAM, DINO) and Task-IDs on 10-Split-CIFAR100 and 5-Split-CUB200.

| Method | Pretrained-dataset | Task ID | 10-Split-CIFAR100 | | 5-Split-CUB200 | |
|---|---|---|---|---|---|---|
| | | | ACC(↑) | Forget(↓) | ACC(↑) | Forget(↓) |
| DualPrompt | ImageNet-21K | Prompt ID | 86.50 | 5.77 | 82.02 | 4.23 |
| DualPrompt-PGP | ImageNet-21K | Prompt ID | 86.92 | 5.35 | 82.46 | 3.76 |
| DualPrompt | SAM | Prompt ID | 86.11 | 6.08 | 82.02 | 4.73 |
| DualPrompt | DINO | Prompt ID | 64.18 | 23.81 | 50.88 | 10.10 |
| DualPrompt-**Soft-TF** | ImageNet-21K | Prompt ID | 92.42 | 2.44 | 76.17 | 9.04 |
| DualPrompt-**Soft-TF** (SOTA) | **ImageNet-21K** | **Gradient ID** | **97.87** | **0.21** | **87.93** | **0.66** |
| DualPrompt-**Soft-TF** (SOTA) | **SAM** | **Gradient ID** | **97.87** | **0.21** | **87.93** | **0.66** |
| DualPrompt-**Soft-TF** | DINO | Gradient ID | 84.50 | 12.27 | 69.79 | 10.93 |
| Upper-Bound of Soft-TF | - | - | 93.90 | - | 85.56 | - |

## 5.2 PERFORMANCES

**Performances of Soft-TF on CIL.** We compare our Soft-TransFormers (Soft-TF) with state-of-the-art CIL baselines, as shown in Table 1. Our Soft-TF significantly outperforms all baselines and upper-bounds of Soft-TF, including L2P and DualPrompt, in both accuracy and forgetting measurements. The performance gain of Soft-TF is especially notable in DualPrompt-based learning compared to L2P, suggesting the importance of global prompt-tuning and multi-head attention prompt-tuning in DualPrompt. Additionally, task-identity inference using Gradient-ID is crucial for achieving full fine-tuning results in CIL. To demonstrate the effectiveness of Soft-TF, Figure 2(a) presents a radar chart comparing DualPrompt-PGP and Soft-TransFormers across four benchmark datasets.

**Well-initialized LTH (WLTH) on CIL.** To demonstrate the efficacy of our proposed method on Well-initialized Lottery Ticket Hypothesis (WLTH) backbones, we evaluate our Soft-TransFormers (Soft-TF) by extending two distinct pre-trained models, ViT-DINO and ViT-SAM (Caron et al., 2021; Chen et al., 2021). As shown in Table 2, we tested our method on the 10-Split-CIFAR100 and 5-Split-CUB200 datasets using three pre-trained ViTs: ImageNet-21K, DINO, and SAM, further validating the effectiveness of our approach on non-ImageNet datasets (Krizhevsky et al., 2009; Wah et al., 2011). Surprisingly, when initialized with ImageNet-21K and SAM, DualPrompt-Soft-TF with ImageNet-21K achieved the same performance levels. Moreover, DualPrompt-Soft-TF outperformed all baselines, i.e., DualPrompt-PGP, on both benchmark datasets, indicating that well-initialized weights provide better generalization in continual learning scenarios.

**CLIP on CIL and TIL.** We conduct our experiments on the 10-Split-CIFAR100 dataset under both Class Incremental Learning (CIL) and Task Incremental Learning (TIL) settings, as shown in Table 3. The results demonstrate that CLIP-Prompt-Soft-TF-L[1-12] significantly improves performance in

Table 3: **Comparisons of Soft-TF with baselines based on CLIP model** on 10-Split-CIFAR100. ∗ denotes our reproduced results.

| Method | Task ID | Class Incremental | | Task Incremental | |
|---|---|---|---|---|---|
| | | ACC(↑) | Forget(↓) | ACC(↑) | Forget(↓) |
| CLIP | Prompt ID | 73.76 | 5.60 | 92.69 | 2.34 |
| CLIP-PGP | Prompt ID | 79.47 | 4.23 | 93.00 | 1.58 |
| CLIP* | Prompt ID | 74.60 | 7.75 | 93.59 | 2.80 |
| CLIP-PGP* | Prompt ID | 74.63 | 7.76 | 93.67 | 2.83 |
| CLIP-Prompt | Prompt ID | 70.27 | 12.95 | 93.36 | 3.07 |
| CLIP-Prompt-**Soft-TF**-L[3,4,5] | Prompt ID | 71.58 | 7.73 | 95.29 | 1.12 |
| CLIP-Prompt-**Soft-TF**-L[3,4,5] | Gradient ID | 76.77 | 5.59 | 95.29 | 1.12 |
| CLIP-Prompt-**Soft-TF**-L[1-12] | Prompt ID | 72.28 | 3.44 | 96.83 | 0.44 |
| CLIP-Prompt-**Soft-TF**-L[1-12] (SOTA) | Gradient ID | **85.90** | **3.07** | **96.83** | **0.44** |

both settings, indicating that our Soft-TF with Gradient ID is also effective in vision-language models, thereby broadening its applicability.

## 5.3 ABLATION STUDIES

**Layer-wise Inspections.** We analyze the layer-wise performance of Soft-Transformer with respect to L2P and DualPrompt on the 10-Split-CIFAR100 dataset to identify the optimal configurations, as shown in Figure 4. Our observations reveal that the global prompt in DualPrompt influences Soft-Transformer's performance differently in L2P and DualPrompt settings. In L2P-PGP, the best performance was achieved with Soft-TransFormers applied to the lower layers ((a) L2P-Soft-TF-L[1,2]-PGP), whereas in DualPrompt, the higher layers ((b) DualPrompt-Soft-TF-L[10,11,12]) yielded the best results. Notably, DualPrompt-Soft-TF-L[10,11,12] without PGP demonstrated impressive performance, achieving almost zero forgetting (0.21). These findings suggest that our approach could significantly enhance the effectiveness of large-scale Transformer models in continual learning scenarios.

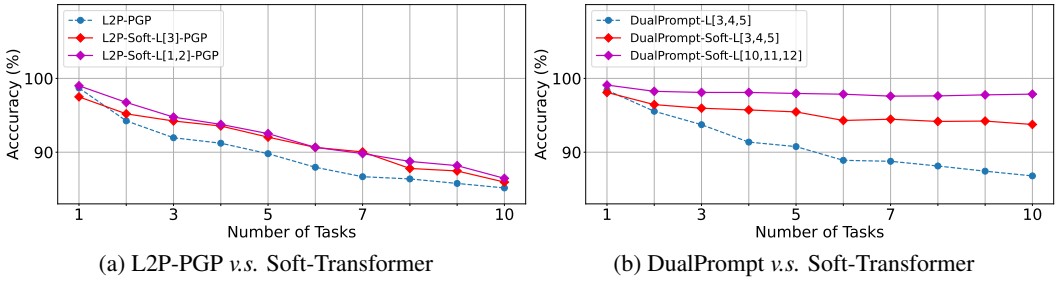

(a) L2P-PGP *v.s.* Soft-Transformer          (b) DualPrompt *v.s.* Soft-Transformer

Figure 4: **Layer-wise(L[∗]) Performances of Soft-TF** on 10-Split-CIFAR100. Note that L[9,10,11] denotes Soft-TransFormer of 9, 10, 11 Layers.

## 6 CONCLUSION

Inspired by *Well-initialized Lottery Ticket Hypothesis (WLTH)* that provides suboptimal fine-tuning solutions, we proposed a novel fully fine-tuned continual learning (CL) method referred to as *Soft-TransFormers (Soft-TF)*, which sequentially learns and selects an optimal soft-network or subnetwork for each task. In training, Soft-TF jointly learned the sparse layer's weights in CL to obtain task-adaptive soft(real-valued)-networks or subnetworks (binary masks) while freezing the well-pre-trained layer parameters. In inference, the identified task-adaptive network of Soft-TF, which masks the parameters of the pre-trained network, maps to an optimal solution associated with each task, minimizing Catastrophic Forgetting (CF)—the soft masking was immune to the pre-trained network's knowledge forgetting. Extensive experiments demonstrated the power of Soft-TF (Vision Transformer and CLIP) and show state-of-the-art performances with convergence theory in various CL scenarios, i.e., Class-Incremental Learning (CIL) and Task-Incremental Learning (TIL).

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

## A  APPENDIX

### A.1  ANALYSIS OF SOFT-TRANSFORMERS (SOFT-TF)

**Analysis of Soft-TransFormers for Convex-Lipschitz Functions**. To analyze the convergence rate of the Soft-TransFormers (Soft-TF), we limit ourselves to the case of convex-Lipschitz functions along with the analysis (Shalev-Shwartz & Ben-David, 2014). Let $\boldsymbol{w}^* = \{\boldsymbol{g}^*, \boldsymbol{e}_t^*, \boldsymbol{m}_t^*\}$ be any vector or an optimal solution and let $B$ be an upper bound on $||\boldsymbol{w}^*||$ when $\boldsymbol{w}^{(1)} = \boldsymbol{0}$. It is convenient to think of $\boldsymbol{w}^*$ as the minimizer of $f(\boldsymbol{w})$, but the analysis that follows holds for every $\boldsymbol{w}^*$.

We would like to obtain an upper bound on the sub-optimality of our solution with respect to $\boldsymbol{w}^*$, namely, $f(\bar{\boldsymbol{w}}) - f(\boldsymbol{w}^*)$, where $\bar{\boldsymbol{w}} = \frac{1}{T}\boldsymbol{w}^{(t)}$. From the definition of $\bar{\boldsymbol{w}}$, and using Jensen's inequality, we have that

$$
\begin{aligned}
f(\bar{\boldsymbol{w}}) - f(\boldsymbol{w}^*) &= f\left(\frac{1}{T}\sum_{t=1}^{T}\boldsymbol{w}^{(t)}\right) - f(\boldsymbol{w}^*) \\
&\leq \frac{1}{T}\sum_{t=1}^{T}\left(f(\boldsymbol{w}^{(t)})\right) - f(\boldsymbol{w}^*) \\
&= \frac{1}{T}\sum_{t=1}^{T}\left(f(\boldsymbol{w}^{(t)}) - f(\boldsymbol{w}^*)\right).
\end{aligned}
\tag{14}
$$

For every $t$, because of the convexity of $f$, we have that

$$
f(\boldsymbol{w}^{(t)}) - f(\boldsymbol{w}^*) \leq \left\langle \boldsymbol{w}^{(t)} - \boldsymbol{w}^*, \nabla f(\boldsymbol{w}^{(t)}) \right\rangle
\tag{15}
$$

Combining the preceeding we obtain

$$
f(\boldsymbol{w}^{(t)}) - f(\boldsymbol{w}^*) \leq \frac{1}{T}\sum_{t=1}^{T}\left\langle \boldsymbol{w}^{(t)} - \boldsymbol{w}^*, \nabla f(\boldsymbol{w}^{(t)}) \right\rangle
\tag{16}
$$

To bound the right-hand side we rely on the following lemma:

**Lemma A.1.** *Let* $\boldsymbol{v}_1, \cdots, \boldsymbol{v}_T$ *be an arbitrary sequence of vectors. Any algorithm with an well initialization (pre-trained model)* $\boldsymbol{w}^{(1)} \neq \boldsymbol{0}$ *and an update rule of the form*

$$
\boldsymbol{w}^{(t+1)} = \boldsymbol{w}^{(t)} - \eta\boldsymbol{v}_t
\tag{17}
$$

*satisfies with $||\boldsymbol{w}^{(1)} - \boldsymbol{w}^*||^2 = ||\boldsymbol{w}^{(T+1)} - \boldsymbol{w}^*||^2$*

$$\sum_{t=1}^{T} \left\langle \boldsymbol{w}^{(t)} - \boldsymbol{w}^*, \boldsymbol{v}_t \right\rangle \leq \frac{1}{2\eta} ||\boldsymbol{w}_m^{(T+1)} - \boldsymbol{w}^*||^2 + \frac{\eta}{2} \sum_{t=1}^{T} ||\boldsymbol{v}_t||^2$$

$$< \frac{1}{2\eta} ||\boldsymbol{w}_p^{(T+1)} - \boldsymbol{w}^*||^2 + \frac{\eta}{2} \sum_{t=1}^{T} ||\boldsymbol{v}_t||^2 \qquad (18)$$

$$< \frac{1}{2\eta} ||\boldsymbol{w}^{(T+1)} - \boldsymbol{w}^*||^2 + \frac{\eta}{2} \sum_{t=1}^{T} ||\boldsymbol{v}_t||^2$$

*where $\boldsymbol{w}_m \neq \boldsymbol{w}_q$ since $\boldsymbol{m}$ is learnable parameters in Soft-TransFormers. Specifically, we could assume that $\boldsymbol{w}_m = (\boldsymbol{w}^Q \odot \boldsymbol{m}^Q) \cdot \boldsymbol{x}\boldsymbol{p}^T \cdot (\boldsymbol{w}^K \odot \boldsymbol{m}^K)^T$ and $\boldsymbol{w}_p = (\boldsymbol{w}^Q \odot \mathbf{1}^Q) \cdot \boldsymbol{x}\boldsymbol{p}^T \cdot (\boldsymbol{w}^K \odot \mathbf{1}^K)^T$ of Equation 6. For every $B_m < B_p < B, \rho > 0$ where $B_m = ||\boldsymbol{w}_m^{(T+1)} - \boldsymbol{w}^*||$ and $B_p = ||\boldsymbol{w}_p^{(T+1)} - \boldsymbol{w}^*||$, if for all $t$ we have that $||\boldsymbol{v}_t \leq \rho||$ and if we set $\eta = \sqrt{\frac{B^2}{\rho^2 T}}$, then for every $\boldsymbol{w}^*$ with $||\boldsymbol{w}^{(T+1)} - \boldsymbol{w}^*|| \leq B$ we have*

$$\frac{1}{T} \sum_{t=1}^{T} \left\langle \boldsymbol{w}^{(t)} - \boldsymbol{w}^*, \boldsymbol{v}_t \right\rangle \leq \frac{B_m \rho}{\sqrt{T}} < \frac{B_p \rho}{\sqrt{T}} < \frac{B\rho}{\sqrt{T}}. \qquad (19)$$

*Proof.* Using algebraic manipulations (completing the square), we obtain:

$$\left\langle \boldsymbol{w}^{(t)} - \boldsymbol{w}^*, \boldsymbol{v}_t \right\rangle = \frac{1}{\eta} \left\langle \boldsymbol{w}^{(t)} - \boldsymbol{w}^*, \eta \boldsymbol{v}_t \right\rangle$$

$$= \frac{1}{2\eta} \left( -||\boldsymbol{w}^{(t)} - \boldsymbol{w}^* - \eta \boldsymbol{v}_t||^2 + ||\boldsymbol{w}^{(t)} - \boldsymbol{w}^*||^2 + \eta^2 ||\boldsymbol{v}_t||^2 \right) \qquad (20)$$

$$= \frac{1}{2\eta} \left( -||\boldsymbol{w}^{(t+1)} - \eta \boldsymbol{v}_t||^2 + ||\boldsymbol{w}^{(t)} - \boldsymbol{w}^*||^2 \right) + \frac{\eta}{2} ||\boldsymbol{v}_t||^2,$$

where the last equality follows from the definition of the update rule. Summing the equality over $t$, we have

$$\sum_{t=1}^{T} \left\langle \boldsymbol{w}^{(t)} - \boldsymbol{w}^*, \boldsymbol{v}_t \right\rangle = \frac{1}{2\eta} \sum_{t=1}^{T} \left( -||\boldsymbol{w}^{(t+1)} - \eta \boldsymbol{v}_t||^2 + ||\boldsymbol{w}^{(t)} - \boldsymbol{w}^*||^2 \right) + \frac{\eta}{2} \sum_{t=1}^{T} ||\boldsymbol{v}_t||^2 \qquad (21)$$

The first sum on the right-hand side is a telescopic sum that collapses to

$$||\boldsymbol{w}^{(1)} - \boldsymbol{w}^*||^2 = ||\boldsymbol{w}^{(T+1)} - \boldsymbol{w}^*||^2 \qquad (22)$$

Plugging this in Equation, we have

$$\sum_{t=1}^{T} \left\langle \boldsymbol{w}^{(t)} - \boldsymbol{w}^*, \boldsymbol{v}_t \right\rangle = \frac{1}{2\eta} \sum_{t=1}^{T} \left( -||\boldsymbol{w}^{(t+1)} - \eta \boldsymbol{v}_t||^2 + ||\boldsymbol{w}^{(t)} - \boldsymbol{w}^*||^2 \right) + \frac{\eta}{2} \sum_{t=1}^{T} ||\boldsymbol{v}_t||^2$$

$$\leq \frac{1}{2\eta} ||\boldsymbol{w}^{(1)} - \boldsymbol{w}^*||^2 + \frac{\eta}{2} \sum_{t=1}^{t} ||\boldsymbol{v}_t||^2 \qquad (23)$$

$$= \frac{1}{2\eta} ||\boldsymbol{w}^*||^2 + \frac{\eta}{2} \sum_{t=1}^{T} ||\boldsymbol{v}_t||^2,$$

where the last equality is due to the definition $\boldsymbol{w}^{(1)} = \mathbf{0}$. This proves the first part of the lemma. The second part follows by upper bounding $||\boldsymbol{w}||$ by $B$, $||\boldsymbol{v}_t||$ by $\rho$, deciding by $T$, and plugging in the value of $\eta$.

In terms of Soft-TransFormers $\boldsymbol{w}_m = (\boldsymbol{w}^Q \odot \boldsymbol{m}^Q) \cdot \boldsymbol{x}\boldsymbol{p}^T \cdot (\boldsymbol{w}^K \odot \boldsymbol{m}^K)^T$ of Equation 6, we have

$$\sum_{t=1}^{T} \left\langle \boldsymbol{w}^{(t)} - \boldsymbol{w}^*, \boldsymbol{v}_t^s \right\rangle = \frac{1}{2\eta} \sum_{t=1}^{T} \left( -||\boldsymbol{w}^{(t+1)} - \eta \boldsymbol{w}_t||^2 + ||\boldsymbol{w}^{(t)} - \boldsymbol{w}^*||^2 \right) + \frac{\eta}{2} \sum_{t=1}^{T} ||\boldsymbol{v}_t||^2$$

$$\leq \frac{1}{2\eta} ||\boldsymbol{w}^{(1)} - \boldsymbol{w}^*||^2 + \frac{\eta}{2} \sum_{t=1}^{t} ||\boldsymbol{v}_t||^2 \qquad (24)$$

where $\boldsymbol{v}_t$ is an arbitrary $t$-th vector and $\boldsymbol{w}^{(1)} \neq \boldsymbol{0}$ since $\boldsymbol{w}^Q$ and $\boldsymbol{q}^K$ are pre-trained parameters.

however, in term of prompt $\boldsymbol{w}_p = (\boldsymbol{w}^Q \odot \mathbf{1}^Q) \cdot \boldsymbol{xp}^T \cdot (\boldsymbol{w}^K \odot \mathbf{1}^K)^T$, we have

$$\sum_{t=1}^{T} \left\langle \boldsymbol{w}^{(t)} - \boldsymbol{w}^*, \boldsymbol{v}_t^p \right\rangle = \frac{1}{2\eta} \sum_{t=1}^{T} \left( -||\boldsymbol{w}^{(t+1)} - \eta \boldsymbol{w}_t||^2 + ||\boldsymbol{w}^{(t)} - \boldsymbol{w}^*||^2 \right) + \frac{\eta}{2} \sum_{t=1}^{T} ||\boldsymbol{v}_t||^2$$

$$\leq \frac{1}{2\eta} ||\boldsymbol{w}^{(1)} - \boldsymbol{w}^*||^2 + \frac{\eta}{2} \sum_{t=1}^{t} ||\boldsymbol{v}_t||^2 \qquad (25)$$

where $\boldsymbol{v}_t$ is an arbitrary $t$-th vector of prompt and $\boldsymbol{w}^{(1)} \neq \boldsymbol{0}$ since $(\boldsymbol{w} \odot \mathbf{1})^Q$ and $(\boldsymbol{w} \odot \mathbf{1})^K$ are pre-trained parameters, $\boldsymbol{w}^Q$ and $\boldsymbol{w}^K$, respectively.

Therefore, we have from $||\boldsymbol{w}^{(1)} - \boldsymbol{w}^*||^2 = ||\boldsymbol{w}^{(T+1)} - \boldsymbol{w}^*||^2$

$$\sum_{t=1}^{T} \left\langle \boldsymbol{w}^{(t)} - \boldsymbol{w}^*, \boldsymbol{v}_t^p \right\rangle \leq \frac{1}{2\eta_m} ||\boldsymbol{w}_m^{(T+1)} - \boldsymbol{w}^*||^2 + \frac{\eta_m}{2} \sum_{t=1}^{t} ||\boldsymbol{v}_t||^2$$

$$< \frac{1}{2\eta_p} ||\boldsymbol{w}_p^{(T+1)} - \boldsymbol{w}^*||^2 + \frac{\eta_p}{2} \sum_{t=1}^{t} ||\boldsymbol{v}_t||^2 \qquad (26)$$

$$< \frac{1}{2\eta} ||\boldsymbol{w}^{(T+1)} - \boldsymbol{w}^*||^2 + \frac{\eta}{2} \sum_{t=1}^{T} ||\boldsymbol{v}_t||^2$$

where $||\boldsymbol{w}_m^{(1)} - \boldsymbol{w}^*||^2 < ||\boldsymbol{w}_p^{(1)} - \boldsymbol{w}^*||^2$ since all $\boldsymbol{m}$ are learnable parameters. For every $B_m < B_p < B, \rho > 0$ where $B_m = ||\boldsymbol{w}_m^{(T+1)} - \boldsymbol{w}^*||$ and $B_p = ||\boldsymbol{w}_p^{(T+1)} - \boldsymbol{w}^*||$, if for all $t$ we have that $||\boldsymbol{v}_t \leq \rho||$ and if we set $\eta \approx \eta_m \approx \eta_p = \sqrt{\frac{B^2}{\rho^2 T}}$ with large enough $T$, then for every $\boldsymbol{w}^*$ with $||\boldsymbol{w}^{(T+1)} - \boldsymbol{w}^*|| \leq B$ we have

$$\frac{1}{T} \sum_{t=1}^{T} \left\langle \boldsymbol{w}^{(t)} - \boldsymbol{w}^*, \boldsymbol{v}_t \right\rangle \leq \frac{B_m \rho}{\sqrt{T}} < \frac{B_p \rho}{\sqrt{T}} < \frac{B \rho}{\sqrt{T}}. \qquad (27)$$

$\square$

## A.2 EXPERIMENTAL DETAILS

For fair comparisons with the baselines (Wang et al., 2022c;b; Qiao et al., 2024), we use ViT B/16 (Dosovitskiy et al., 2020) pre-trained on ImageNet-21K as our image encoder, which is kept frozen during training. We train and test on a single Quadro RTX 8000-48GB GPU for baselines and our Soft-TransFormers with Adam optimizer with $\beta_1 = 0.9$ and $\beta_2 = 0.999$.

We adhere to the experimental settings outlined by Qiao et al. (2024) to validate our method's effectiveness. When comparing our approach with L2P-PGP and Soft-Transformer on the 10/20-Split-CIFAR100 and 10-Split-TinyImageNet datasets, we train the network for 5 epochs with a batch size of 16 and set the prompt length to 5. For the 10-Split-ImageNet-R dataset, we use 50 epochs, a batch size of 16, and a prompt length of 30. In comparison with DualPrompt-PGP and Soft-TransFormers on the 10/20-Split-CIFAR100 dataset, we train the network for 20 epochs with a batch size of 24 and set the expert prompt length to 5. For the 10-Split-TinyImageNet dataset, we use 5 epochs, a batch

size of 24, and an expert prompt length of 5. For the 10-Split-ImageNet-R dataset, we set the epochs to 50, the batch size to 24, and the expert prompt length to 20. Additionally, in all benchmark data sets, the general prompt length is set to 5, and the location inserted into the prompt is kept consistent.

For CLIP-PGP and Soft-TransFormers, we configure a single trainable image prompt that is shared across all tasks within the vision encoder. For the text encoder, following the approach of Qiao et al. (2024), we set a trainable text prompt for each class, which is only trained on the corresponding task. In our comparisons with CLIP-PGP and Soft-TransFormers on the 10-Split-CIFAR100 dataset, we set the image prompt length to 5, the number of epochs to 5, and the batch size to 32.

Table 4: **Performances of Class Incremental Learning (CIL)** in terms of accuracy and forgetting on 10/20-Split-CIFAR100 and 10-Split-ImageNet-R. Exemplar means the total buffer size for rehearsal methods.

| Method | Exemplar | Task ID | 10-Split-CIFAR100 | | 20-Split-CIFAR100 | | 10-Split-ImageNet-R | |
| | | | ACC(↑) | Forget(↓) | ACC(↑) | Forget(↓) | ACC(↑) | Forget(↓) |
|---|---|---|---|---|---|---|---|---|
| BiC | 5,000 | - | 81.42 | 17.31 | 73.02 | 6.23 | 64.63 | 22.25 |
| DER++ | 5,000 | - | 83.94 | 14.55 | - | - | 66.73 | 20.67 |
| iCaRL | 5,000 | - | 66.00 | 5.33 | 78.02 | 5.80 | - | - |
| DER+MCG | 2,000 | - | 67.62 | 14.64 | 65.84 | 13.72 | - | - |
| BiC | 1,000 | - | 66.11 | 35.24 | 63.12 | 21.89 | 52.14 | 36.70 |
| DER++ | 1,000 | - | 61.06 | 39.87 | - | - | 55.47 | 34.64 |
| iCaRL | 1,000 | - | 61.25 | 14.19 | 71.32 | 15.98 | - | - |
| FT | - | - | 33.61 | 86.87 | 33.52 | 53.69 | 28.87 | 63.80 |
| EWC | - | - | 47.01 | 33.27 | 36.73 | 35.19 | 35.00 | 56.16 |
| LWF | - | - | 60.69 | 27.77 | 39.12 | 57.91 | 38.54 | 52.37 |
| L2P* | - | Prompt ID | 83.77 | 6.63 | 71.29 | 13.96 | 60.44 | 9.00 |
| L2P-PGP* | - | Prompt ID | 84.34 | 5.59 | 76.12 | **13.26** | 61.40 | 8.03 |
| L2P-PGP-**Soft-TF** | - | Prompt ID | 86.26 | **4.79** | 76.17 | 15.77 | **69.80** | **5.13** |
| L2P-PGP-**Soft-TF** | - | Gradient ID | **86.46** | 4.87 | **77.67** | 15.84 | 69.56 | 5.28 |
| DualPrompt | - | Prompt ID | 86.50 | 5.77 | 82.98 | 8.20 | 68.13 | 4.46 |
| DualPrompt-**Soft-TF**-L[3,4,5] | - | Prompt ID | 91.77 | 3.37 | 94.43 | 2.02 | 74.70 | 6.46 |
| DualPrompt-**Soft-TF**-L[3,4,5] | - | Gradient ID | 93.76 | 1.83 | 95.38 | 1.73 | 82.15 | 2.20 |
| DualPrompt-**WSN**-L[10,11,12], c=80.0% | - | Gradient ID | 97.41 | 0.18 | 90.25 | 9.08 | 74.83 | 0.91 |
| DualPrompt-**WSN**-L[10,11,12], c=81.0% | - | Gradient ID | 97.50 | 0.21 | 96.72 | 2.21 | 74.21 | 1.41 |
| DualPrompt-**WSN**-L[10,11,12], c=82.0% | - | Gradient ID | 97.67 | 0.27 | 96.44 | 1.62 | 75.02 | 0.92 |
| DualPrompt-**WSN**-L[10,11,12], c=83.0% | - | Gradient ID | 97.62 | 0.25 | 97.77 | 0.63 | 76.33 | 1.77 |
| DualPrompt-**WSN**-L[10,11,12], c=87.0% | - | Gradient ID | 97.51 | 0.27 | 97.68 | 0.75 | 77.96 | 1.02 |
| DualPrompt-**WSN**-L[10,11,12], c=90.0% | - | Gradient ID | 97.46 | 0.38 | 98.09 | 0.65 | 78.80 | 0.47 |
| DualPrompt-**Soft-TF**-L[10,11,12] | - | Gradient ID | **97.87** | **0.21** | **99.05** | **0.24** | **82.38** | **0.59** |
| DualPrompt-PGP | - | Prompt ID | 86.92 | 5.35 | 83.74 | 7.91 | 69.34 | 4.53 |
| DualPrompt-PGP-**Soft-TF**-L[3,4,5] | - | Prompt ID | 92.41 | 2.44 | 95.14 | 1.90 | 74.65 | 4.39 |
| DualPrompt-PGP-**Soft-TF**-L[3,4,5] | - | Gradient ID | **92.92** | **2.34** | **95.89** | **1.64** | **81.45** | **2.89** |
| Upper-Bound of DualPrompt | - | - | 90.85 | - | 90.85 | - | 79.13 | - |
| Upper-Bound of Soft-TF | - | - | 93.90 | - | 93.90 | - | 80.21 | - |

Table 5: **Random initialized Performances of Class Incremental Learning (CIL)** in terms of accuracy and forgetting on 10-Split-CIFAR100. Note "w/o FF" denotes "Soft fine-tuning without FeedForward (FF)" networks.

| Method | Pretrained-Dataset | Task ID | Random Initialization | 10-Split-CIFAR100 | |
| | | | | ACC(↑) | Forget(↓) |
|---|---|---|---|---|---|
| DualPrompt-**Soft-TF**-L[10,11,12] w/o FF | ImageNet-21K | Prompt ID | Xavier | 90.59 | 3.85 |
| DualPrompt-**Soft-TF**-L[10,11,12] w/o FF | ImageNet-21K | Prompt ID | Kaiming | 90.72 | 3.63 |
| DualPrompt-**Soft-TF**-L[10,11,12] w/o FF | ImageNet-21K | Prompt ID | Normal | 90.45 | 3.78 |
| DualPrompt-**Soft-TF**-L[10,11,12] w/o FF | ImageNet-21K | Prompt ID | **Uniform(1.0, 1.0)** | 92.35 | 2.98 |
| DualPrompt-**Soft-TF**-L[10,11,12] w/o FF | ImageNet-21K | **Gradient ID** | **Uniform(1.0, 1.0)** | **98.05** | **0.25** |
| Upper-Bound of Soft-TF | | | | 93.90 | - |

**Random initialization**. Random initialization of Soft-Transformer's weights plays a critical role when leveraging well-pretrained models like Vision Transformers (ViTs). The optimal training point is the parameters of a well-pretrained model. Among the initialization methods, Uniform initialization for Soft-TransFormer satisfies this requirement effectively. To validate these claims, we analyze the impact of common random initialization methods, including Xavier, Kaiming, Normal, and Uniform Initialization, as shown in Table 5. The results demonstrate that the same well-initialization point leads to independent optimal task performance, particularly with Gradient ID inference. Furthermore, this ablation study strengthens our Soft-TF with state-of-the-art-perfomances inspired by the Well-initialized Lottery Ticket Hypothesis (WLTH).

**Training & Test Time**. To clearly illustrate the time complexity of Soft-TF, we present the training and testing times for 10/20-Split-CIFAR100 and 10-Split-ImageNet-R, as shown in Table 6. As the number of trainable parameters in Soft-TF increases, training and testing time complexities grow

Table 6: **Performances of Class Incremental Learning (CIL)** in terms of **Soft parameters, training, and test time** on 10/20-Split-CIFAR100 and 10-Split-ImageNet-R. Note "w/o FF" denotes "Soft finetuning without FeedForward (FF)" networks.

| Method DualPrompt | ViT-B/12 (85.8M) # Train Params. | Task ID | 10-Split-CIFAR100 Train (sec.) | Test (sec.) | 20-Split-CIFAR100 Train (sec.) | Test (sec.) | 10-Split-ImageNet-R Train (sec.) | Test (sec.) |
|---|---|---|---|---|---|---|---|---|
| DualPrompt | 0.00M | Prompt ID | 12.12K | 76 | 11.60K | 78 | 13.10K | 47 |
| **PGP** | 0.00M | Prompt ID | 12.21K | 76 | 13.12K | 78 | 13.33K | 47 |
| **Soft-TF**-L[12] w/ only ATTN | 1.76M | Gradient ID | 12.18K | 129 | 13.30K | 113 | 13.35K | 65 |
| **Soft-TF**-L[12] w/ only ATTN | 1.76M | Prompt ID | 12.18K | 78 | 13.30K | 80 | 13.35K | 48 |
| **Soft-TF**-L[12] w/o FF | 2.31M | Gradient ID | 12.24K | 103 | 13.40K | 132 | 13.42K | 66 |
| **Soft-TF**-L[11,12] w/o FF | 4.62M | Gradient ID | 12.95K | 115 | 14.38K | 146 | 14.23K | 73 |
| **Soft-TF**-L[10,11,12] w/o FF | 6.93M | Gradient ID | 13.71K | 130 | 15.51K | 163 | 15.08K | 82 |
| **Soft-TF**-L[10,11,12] w/o FF | 6.93M | Prompt ID | 13.87K | 80 | 15.60K | 104 | 15.35K | 52 |
| **LoRA**-L[10,11,12] w/o FF, r=4 | 0.06M | Prompt ID | 11.95K | 77 | 11.71K | 79 | 14.34K | 48 |
| **LoRA**-L[10,11,12] w/o FF, r=24 | 0.32M | Prompt ID | 12.03K | 78 | 15.10K | 100 | 15.02K | 50 |
| **LoRA**-L[10,11,12] w/o FF, r=500 | 6.91M | Prompt ID | 13.24K | 79 | 15.89K | 105 | 15.09K | 53 |
| **Adapter**-L[10,11,12] w/ FF, r=1 | 0.09M | Prompt ID | 12.44K | 84 | 12.40K | 81 | 14.40K | 50 |
| **Adapter**-L[10,11,12] w/ FF, r=4 | 0.36M | Prompt ID | 12.80K | 85 | 15.35K | 105 | 14.68K | 51 |
| **Adapter**-L[10,11,12] w/ FF, r=75 | 6.91M | Prompt ID | 13.66K | 88 | 15.72K | 106 | 15.50K | 53 |

Table 7: **Performances of Class Incremental Learning (CIL)** in terms of **Soft parameters, Accuracy, and Forget** on 10/20-Split-CIFAR100 and 10-Split-ImageNet-R. Note "w/o FF" denotes "Soft finetuning without FeedForward (FF)" networks.

| Method DualPrompt | ViT-B/12 (85.8M) # Train Params. | Task ID | 10-Split-CIFAR100 ACC(↑) | Forget(↓) | 20-Split-CIFAR100 ACC(↑) | Forget(↓) | 10-Split-ImageNet-R ACC(↑) | Forget(↓) |
|---|---|---|---|---|---|---|---|---|
| DualPrompt | 0.00M | Prompt ID | 86.50 | 5.77 | 82.98 | 8.20 | 68.13 | 4.46 |
| **PGP** | 0.00M | Prompt ID | 86.92 | 5.35 | 83.74 | 7.91 | 69.34 | 4.53 |
| **Soft-TF**-L[12] w/ only ATTN | 1.76M | Gradient ID | 97.17 | 0.40 | 98.09 | 0.54 | 72.31 | 3.94 |
| **Soft-TF**-L[12] w/ only ATTN | 1.76M | Prompt ID | 94.59 | 1.12 | 96.96 | 1.02 | 71.13 | 4.93 |
| **Soft-TF**-L[12] w/o FF | 2.31M | Gradient ID | 96.84 | 0.55 | 97.81 | 0.57 | 81.18 | 1.31 |
| **Soft-TF**-L[11,12] w/o FF | 4.62M | Gradient ID | 97.58 | 0.34 | 98.65 | 0.43 | 83.09 | 0.42 |
| **Soft-TF**-L[10,11,12] w/o FF | 6.93M | Gradient ID | 98.05 | 0.25 | 98.96 | 0.23 | **83.70** | **0.53** |
| **Soft-TF**-L[10,11,12] w/o FF | 6.93M | Prompt ID | 92.35 | 2.98 | 97.40 | 0.57 | 76.62 | 5.30 |
| **LoRA**-L[10,11,12] w/o FF, r=4 | 0.06M | Prompt ID | 82.19 | 4.33 | 93.74 | 2.07 | 70.91 | 9.11 |
| **LoRA**-L[10,11,12] w/o FF, r=24 | 0.32M | Prompt ID | 86.77 | 4.27 | 95.65 | 1.04 | 69.81 | 10.30 |
| **LoRA**-L[10,11,12] w/o FF, r=500 | 6.91M | Prompt ID | 82.00 | 4.33 | 92.14 | 2.02 | 43.51 | 13.21 |
| **Adapter**-L[10,11,12] w/ FF, r=1 | 0.09M | Prompt ID | 86.38 | 4.87 | 85.61 | 5.04 | 70.95 | 4.31 |
| **Adapter**-L[10,11,12] w/ FF, r=4 | 0.36M | Prompt ID | 86.53 | 4.52 | 85.66 | 5.00 | 70.82 | 4.90 |
| **Adapter**-L[10,11,12] w/ FF, r=75 | 6.91M | Prompt ID | 86.45 | 4.61 | 84.75 | 5.11 | 70.55 | 4.74 |

accordingly. While the testing time complexity of Gradient ID increased by approximately 1.6 times across the three benchmark datasets, it consistently improved task performance on all benchmarks. The corresponding performance metrics are detailed in Table 7.

We investigate the most parameter-efficient and gradient-based task inference methods, as shown in Table 8 and Table 9. Our findings reveal that the 3-shot Gradient ID inference cost (using samples within a mini-batch) with the last layer (Soft-TF-L[12]) is approximately 1.1 times that of Prompt ID, maintaining comparable efficiency while delivering superior performance. Note that m-batch denotes mini-batch.

Table 8: **Performances of Class Incremental Learning (CIL)** in terms of **Soft parameters, training, and test time** on 10/20-Split-CIFAR100 and 10-Split-ImageNet-R. Note "w/o FF" denotes "Soft finetuning without FeedForward (FF)" networks.

| Method DualPrompt | ViT-B/12 (85.8M) # Train Params. | Task ID | 10-Split-CIFAR100 Train (sec.) | Test (sec.) | 20-Split-CIFAR100 Train (sec.) | Test (sec.) | 10-Split-ImageNet-R Train (sec.) | Test (sec.) |
|---|---|---|---|---|---|---|---|---|
| DualPrompt | 0.00M | Prompt ID | 12.12K | 76 | 11.60K | 78 | 13.10K | 47 |
| **PGP** | 0.00M | Prompt ID | 12.21K | 76 | 13.12K | 78 | 13.33K | 47 |
| **Soft-TF**-L[12] w/o FF | 2.31M | Prompt ID | 12.24K | 79 | 13.40K | 80 | 13.42K | 48 |
| **Soft-TF**-L[12] w/o FF | 2.31M | Gradient ID, 3-shot | 12.24K | 88 | 13.40K | 90 | 13.42K | 57 |
| **Soft-TF**-L[12] w/o FF | 2.31M | Gradient ID, 5-shot | 12.24K | 94 | 13.40K | 98 | 13.42K | 61 |
| **Soft-TF**-L[12] w/o FF | 2.31M | Gradient ID, 7-shot | 12.24K | 95 | 13.40K | 108 | 13.42K | 62 |
| **Soft-TF**-L[12] w/o FF | 2.31M | Gradient ID, m-batch | 12.24K | 103 | 13.40K | 132 | 13.42K | 66 |
| **Soft-TF**-L[10,11,12] w/o FF | 6.93M | Prompt ID | 13.87K | 80 | 15.60K | 104 | 15.35K | 52 |
| **Soft-TF**-L[10,11,12] w/o FF | 6.93M | Gradient ID, 3-shot | 13.71K | 96 | 15.51K | 106 | 15.08K | 63 |
| **Soft-TF**-L[10,11,12] w/o FF | 6.93M | Gradient ID, 5-shot | 13.71K | 96 | 15.51K | 109 | 15.08K | 74 |
| **Soft-TF**-L[10,11,12] w/o FF | 6.93M | Gradient ID, 7-shot | 13.71K | 106 | 15.51K | 119 | 15.08K | 75 |
| **Soft-TF**-L[10,11,12] w/o FF | 6.93M | Gradient ID, batch | 13.71K | 130 | 15.51K | 163 | 15.08K | 82 |

**Comparisions of Soft-TF with LLMs**. To demonstrate the effectiveness of Soft-TF, we compare Soft-TF against LLM fine-tuning methods such as Adapters (Houlsby et al., 2019) and LoRA (Hu

Table 9: **Performances of Class Incremental Learning (CIL)** in terms of **Soft parameters, Accuracy, and Forget** on 10/20-Split-CIFAR100 and 10-Split-ImageNet-R. Note "w/o FF" denotes "Soft finetuning without FeedForward (FF)" networks.

| Method
DualPrompt | ViT-B/12 (85.8M)
# Train Params. | Task ID | 10-Split-CIFAR100
ACC(↑) | Forget(↓) | 20-Split-CIFAR100
ACC(↑) | Forget(↓) | 10-Split-ImageNet-R
ACC(↑) | Forget(↓) |
|---|---|---|---|---|---|---|---|---|
| DualPrompt | 0.00M | Prompt ID | 86.50 | 5.77 | 82.98 | 8.20 | 68.13 | 4.46 |
| **PGP** | 0.00M | Prompt ID | 86.92 | 5.35 | 83.74 | 7.91 | 69.34 | 4.53 |
| **Soft-TF**-L[12] w/o FF | 2.31M | Prompt ID | 91.83 | 2.99 | 96.43 | 1.00 | 72.45 | 5.32 |
| **Soft-TF**-L[12] w/o FF | 2.31M | Gradient ID, 3-shot | 93.12 | 1.82 | 96.43 | 1.00 | 73.55 | 4.80 |
| **Soft-TF**-L[12] w/o FF | 2.31M | Gradient ID, 5-shot | 96.13 | 0.58 | 96.43 | 1.00 | 75.04 | 4.49 |
| **Soft-TF**-L[12] w/o FF | 2.31M | Gradient ID, 7-shot | 96.51 | 0.65 | 96.43 | 1.00 | 76.34 | 4.75 |
| **Soft-TF**-L[12] w/o FF | 2.31M | Gradient ID, batch | 96.84 | 0.55 | 97.81 | 0.57 | 81.18 | 1.31 |
| **Soft-TF**-L[10,11,12] w/o FF | 6.93M | Prompt ID | 92.35 | 2.98 | 97.40 | 0.57 | 74.62 | 5.30 |
| **Soft-TF**-L[10,11,12] w/o FF | 6.93M | Gradient ID, 3-shot | 93.92 | 1.62 | 97.40 | 0.57 | 74.99 | 4.21 |
| **Soft-TF**-L[10,11,12] w/o FF | 6.93M | Gradient ID, 5-shot | 97.37 | 0.53 | 97.40 | 0.57 | 77.40 | 2.91 |
| **Soft-TF**-L[10,11,12] w/o FF | 6.93M | Gradient ID, 7-shot | 97.76 | 0.51 | 97.40 | 0.57 | 79.33 | 3.57 |
| **Soft-TF**-L[10,11,12] w/o FF | 6.93M | Gradient ID, m-batch | **98.05** | **0.25** | **98.96** | **0.23** | **83.70** | **0.53** |

et al., 2021), as shown in Table 7. Under identical experimental conditions—including trainable model parameters ( 6.9M per task), layers (L[10,11,12]), and Prompt ID—Soft-TF outperformed other LLM-based fine-tuning approaches. The results highlight that directly updating well-pretrained model parameters and prompt-tuning via Soft-TF is more effective than combining representations through LoRA or learning representations with Adapters. Furthermore, we observed that Soft-TF and the other methods exhibited comparable training and testing time complexity for the same number of trainable parameters. Notably, single-layer fine-tuning using Soft-TF (with L[12]) surpasses the performance of the baselines. These findings firmly establish Soft-TF as the most competitive approach among strong LLM baselines (Adapters and LoRA) in the continual learning (CIL) scenario.

**Sparsity of Transformer.** We inspect the sparse solution through WSN as shown in Table 4, Table 10, and Table 11. We found a suboptimal sparse solution (c=87.0 % on 10-Split-TinyImageNet) with minimal CF through the inspections. This demonstrates the Rottary Ticket Hypothesis (RTH) in transformers, a competitive sparse subnetwork in DenseNetwork. In addition, DualPrompt is the lower-bound while DualPrompt-Soft-TF-∗ is the upper-bound, close to the optimal performances.

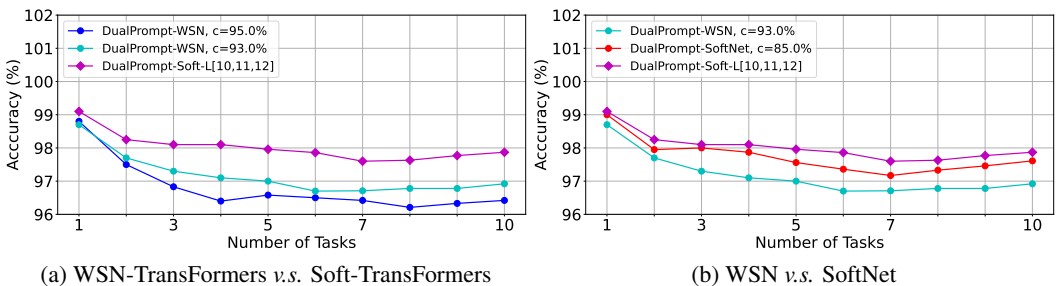

(a) WSN-TransFormers *v.s.* Soft-TransFormers

(b) WSN *v.s.* SoftNet

Figure 5: **Comparisions of Soft-TransFormers with Subnetworks** on 10-Split-CIFAR100. Note that L[10,11,12] denotes the fine-tuning layers of 10, 11, and 12.

Table 10: **Performances of Subnetworks (WSN) in Class Incremental Learning (CIL)** on 10-Split-TinyImageNet.

| Method | Pretrained-Dataset | Task ID | TinyImageNet
ACC(↑) | Forget(↓) |
|---|---|---|---|---|
| DualPrompt | - | Prompt ID | 86.50 | 5.77 |
| DualPrompt-WSN-L[10,11,12] | C=80.0% | Gradient ID | 89.95 | 0.98 |
| DualPrompt-WSN-L[10,11,12] | C=81.0% | Gradient ID | 89.99 | 1.06 |
| DualPrompt-WSN-L[10,11,12] | C=82.0% | Gradient ID | 89.59 | 1.18 |
| DualPrompt-WSN-L[10,11,12] | C=83.0% | Gradient ID | 90.60 | 0.72 |
| DualPrompt-WSN-L[10,11,12] | C=85.0% | Gradient ID | 90.09 | 1.07 |
| DualPrompt-WSN-L[10,11,12] | C=87.0% | Gradient ID | 91.91 | 0.38 |
| DualPrompt-WSN-L[10,11,12] | C=90.0% | Gradient ID | 91.28 | 0.42 |
| DualPrompt-WSN-L[10,11,12] | C=93.0% | Gradient ID | 91.41 | 0.40 |
| DualPrompt-WSN-L[10,11,12] | C=95.0% | Gradient ID | 90.91 | 0.83 |
| DualPrompt-Soft-TF-L[10,11,12] | - | Gradient ID | **97.87** | **0.21** |

Table 11: **Performances of Class Incremental Learning (CIL)** in terms of **Soft parameters, Accuracy, and Forget** on 10/20-Split-CIFAR100 and 10-Split-ImageNet-R. Note "w/ only ATTN" denotes "Soft finetuning only with attention (ATTN)" and FeedForward (FF) networks.

| Method
DualPrompt | ViT-B/12 (85.8M)
# Train Params. | Task ID | 10-Split-CIFAR100 | | 20-Split-CIFAR100 | | 10-Split-ImageNet-R | |
|---|---|---|---|---|---|---|---|---|
| | | | ACC($\uparrow$) | Forget($\downarrow$) | ACC($\uparrow$) | Forget($\downarrow$) | ACC($\uparrow$) | Forget($\downarrow$) |
| DualPrompt | 0.00M | Prompt ID | 86.50 | 5.77 | 82.98 | 8.20 | 68.13 | 4.46 |
| **PGP** | 0.00M | Prompt ID | 86.92 | 5.35 | 83.74 | 7.91 | 69.34 | 4.53 |
| **Soft-TF**-L[12] w/ only ATTN | 1.76M | Gradient ID | 97.17 | 0.40 | 98.09 | 0.54 | 72.31 | 3.94 |
| **Soft-TF**-L[12] w/ only ATTN, WSN c=90% | **1.58M** | Gradient ID | 96.81 | 0.61 | 97.51 | 1.68 | 71.67 | 3.94 |

**Peudo Codes.** The overall process of the Soft-TransFormers (Soft-TF) during training and testing is described as Algorithm 1 and Algorithm 2. We denote the architecture with attached prompts as $f_{\boldsymbol{g}, \boldsymbol{e}_t, \boldsymbol{m}_t}$. The input $\boldsymbol{x}$ from the $t$-th task is transformed using $f_{\boldsymbol{g}, \boldsymbol{e}_t, \boldsymbol{m}_t}$ and then passed to the classification head $f_\phi$, parameterized by $\phi$, for prediction. Finally, we train the two prompts, the task keys, the soft-attention parameters, and the newly-initialized classification head in an end-to-end manner.

---

**Algorithm 1** DualPrompt-Soft-TF at training time

---

1: **Input**: Pre-trained transformer-based backbone $f$, final classification layer $f_\phi$,
2:     number of tasks $\mathcal{T}$, training set $\{\{\boldsymbol{x}_{i,t}, y_{i,t}\}_{i=1}^{n_t}\}_{t=1}^{\mathcal{T}}$, G-Prompt $\boldsymbol{g}$, E-Prompt $\boldsymbol{E} = \{\boldsymbol{e}_t\}_{t=1}^{\mathcal{T}}$,
3:     task keys $\boldsymbol{K} = \{\boldsymbol{k}_t\}_{t=1}^{T}$, soft-networks $\boldsymbol{M} = \{\boldsymbol{m}_t\}_{t=1}^{\mathcal{T}}$ ,$start_g, end_g, start_e, end_e$,
4:     prompting function $f_{\boldsymbol{\theta} \odot \boldsymbol{m}}^{prompt}$,
5:     number of training epochs of the $t$-th task $\mathcal{K}_t$.
6: **Initialize**: $\phi, \boldsymbol{g}, \boldsymbol{E}, \boldsymbol{M}, \boldsymbol{K}$
7: **for** task $t = 1, \cdots, \mathcal{T}$ **do**
8:     Select the task-specific E-Prompt, soft-network $\boldsymbol{e}_t, \boldsymbol{m}_t$ and corresponding task key $\boldsymbol{k}_t$
9:     Generate the prompted architecture $f_{\boldsymbol{g}, \boldsymbol{e}_t, \boldsymbol{m}_t}$: attach $\boldsymbol{g}$ and $\boldsymbol{e}_t$ to $start_g$-th to $end_g$-th
10:       and $start_e$-th to $end_e$-th soft MSA layers respectively, with $f_{\boldsymbol{\theta} \odot \boldsymbol{m}}^{prompt}$.
11:     **for** batch $\boldsymbol{e}_s \sim \mathcal{K}_t$ **do**
12:       Draw a mini-batch $B = \{(\boldsymbol{x}_{i,t}, y_{i,t})\}_{i=1}^l$
13:       **for** $(\boldsymbol{x}, y)$ in B **do**
14:         Calculate the prompted feature by
15:         Calculate the per sample loss $\mathcal{L}_x$ via
16:       **end for**
17:       Update $\phi, \boldsymbol{g}, \boldsymbol{E}, \boldsymbol{M}, \boldsymbol{K}$ by back-propagation
18:     **end for**
19: **end for**

---

**Algorithm 2** DualPrompt-Soft-TF at test time

---

1: **Given components**: Pre-trained transformer-based backbone $f$, trained
2:     $\boldsymbol{K} = \{\boldsymbol{k}_t\}_{t=1}^{\mathcal{T}}, \boldsymbol{M} = \{\boldsymbol{m}_t\}_{t=1}^{\mathcal{T}}, start_g, end_g, start_e, end_e$, prompting function $f_{\boldsymbol{\theta} \odot \boldsymbol{m}}^{prompt}$
3: **Input**: test example $\boldsymbol{x}$ from mini-batch $\boldsymbol{b}$
4: **Select task inference method: (1) Prompt ID or (2) Gradient ID**
5:    **(1) Prompt ID:**
6:     Generate query feature $q(\boldsymbol{x})$
7:     Matching for the index of E-Prompt via $t_{\boldsymbol{x}} = \text{argmin}_t \gamma(q(\boldsymbol{x}), \boldsymbol{k}_t)$
8:    **(2) Gradient ID:**
9:     Assigning each learned subnetwork $\boldsymbol{m}_t$ a weight $\alpha_t$ such that $\sum_t \alpha_t = 1$ and $\alpha_t = 1/\mathcal{T} > 0$.
10:     Given $\boldsymbol{x} \in \boldsymbol{b}$ to classify, we can compute our loss $\mathcal{L} = \mathcal{H}(f_{\boldsymbol{\theta} \odot (\sum_t \alpha_t \boldsymbol{m}_t)}^{prompt}(\boldsymbol{x}))$
11:     Matching for the index of E-Prompt via $t_{\boldsymbol{x}} = \text{argmin}_t \frac{\partial \mathcal{H}}{\partial \alpha_t}$
12: Select the task-specific E-prompt $\boldsymbol{e}_{t_{\boldsymbol{x}}}$ and learned subnetwork $\boldsymbol{m}_{t_{\boldsymbol{x}}}$
13: Generate the prompted architecture $f_{\boldsymbol{g}, \boldsymbol{e}_{t_{\boldsymbol{x}}}, \boldsymbol{m}_{t_{\boldsymbol{x}}}}$:
14:     Attaching $\boldsymbol{g}$ and $\boldsymbol{e}_{t_{\boldsymbol{x}}}$ to $start_g$-th to $end_g$-th
15:     and $start_e$-th to $end_e$-th MSA layers respectively, with $f_{\boldsymbol{\theta} \odot \boldsymbol{m}}^{prompt}$.
16: Prediction: $f_{\boldsymbol{g}, \boldsymbol{e}_{t_{\boldsymbol{x}}}, \boldsymbol{m}_{t_{\boldsymbol{x}}}}(\boldsymbol{x})$

---

**Density of Parameters.** We inspect the histogram density estimate of the last (12) layer's parameters of DualPrompt-Soft-TF: attention of QKV ((a) ATTN.QKV) and Projection ((b) ATTN.PROJ) and multi-layer perception (MLP) of FC1 and FC2, as shown in Figure 6. ATTN's QKV parameters

have the largest variance among the parameter densities, while MLP-FC2's are the smallest. From this observation, we conclude that fine-tuning ATTN's QKV is required to achieve optimal task performance. In other words, QKV's parameters are more critical than others.

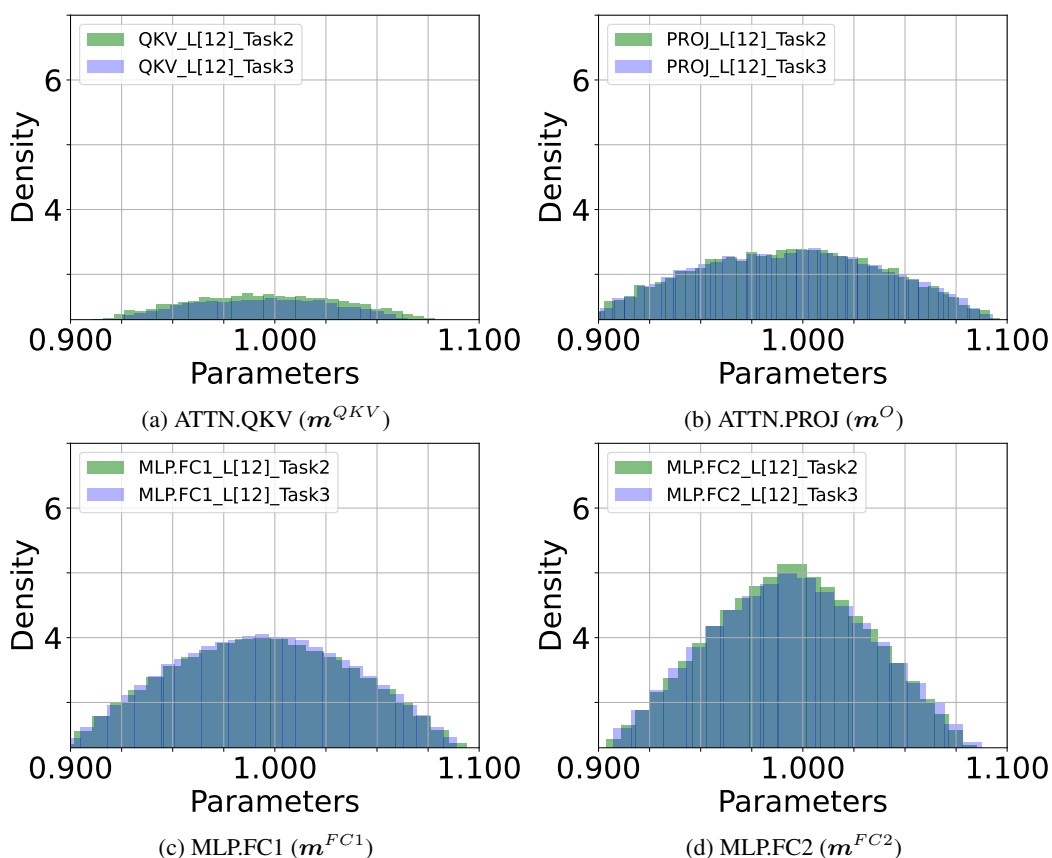

(a) ATTN.QKV ($m^{QKV}$)

(b) ATTN.PROJ ($m^{O}$)

(c) MLP.FC1 ($m^{FC1}$)

(d) MLP.FC2 ($m^{FC2}$)

Figure 6: **Layer-(L[12]) Histogram Density Estimates of DualPrompt-Soft-TF's Parameters** on 10-Split-CIFAR100.

**Pre-trained Parameters v.s. Soft-TF.** We inspect the histogram density estimate of the last (12) layer's parameters of pre-trained model and DualPrompt-Soft-TF: attention of QKV ((a) ATTN.QKV) and Projection ((b) ATTN.PROJ) and multi-layer perception (MLP) of FC1 and FC2, as shown in Figure 7. The most parameters of DualPrompt-Soft-TF are trained around zero-values. Particularly, the difference between pre-trained model's parameters and Soft-TF is distinctive at QKV module.

**Public Source Code.** All official source codes will be available soon.

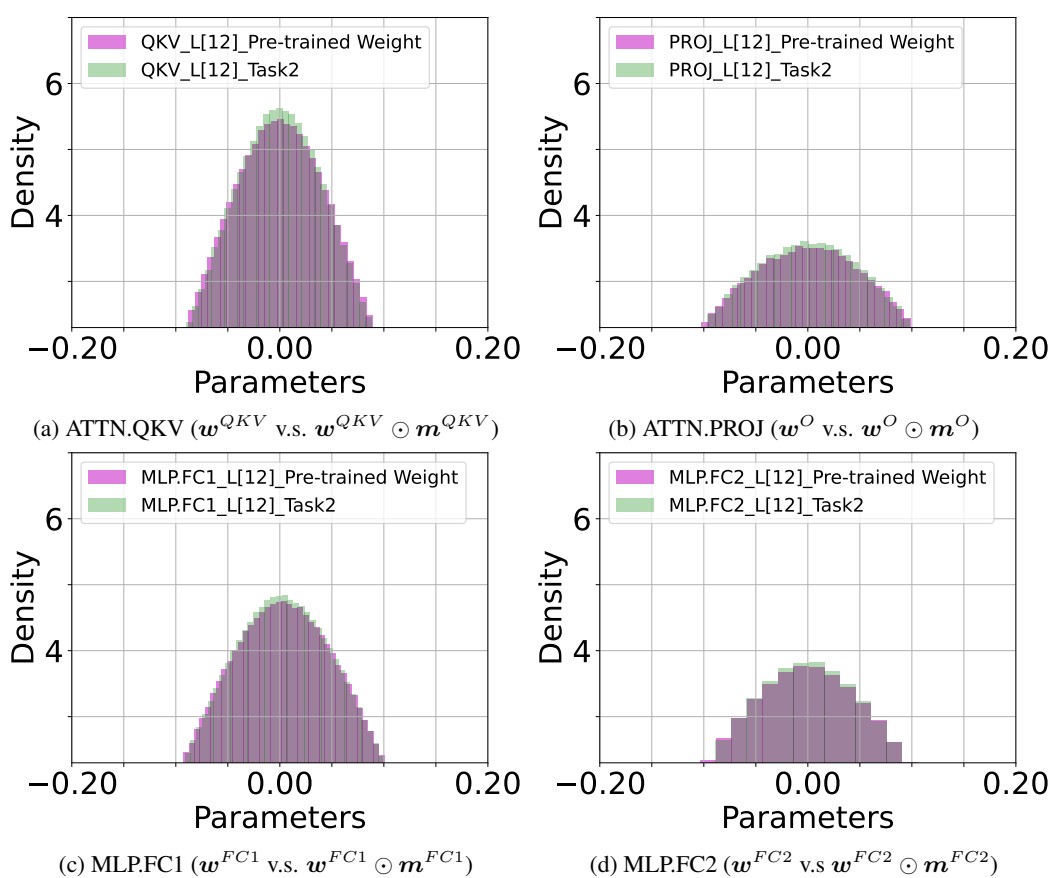

(a) ATTN.QKV ($\boldsymbol{w}^{QKV}$ v.s. $\boldsymbol{w}^{QKV} \odot \boldsymbol{m}^{QKV}$)

(b) ATTN.PROJ ($\boldsymbol{w}^{O}$ v.s. $\boldsymbol{w}^{O} \odot \boldsymbol{m}^{O}$)

(c) MLP.FC1 ($\boldsymbol{w}^{FC1}$ v.s. $\boldsymbol{w}^{FC1} \odot \boldsymbol{m}^{FC1}$)

(d) MLP.FC2 ($\boldsymbol{w}^{FC2}$ v.s $\boldsymbol{w}^{FC2} \odot \boldsymbol{m}^{FC2}$)

Figure 7: **Layer-(L[12]) Histogram Density Estimates of Pre-trained Weight and DualPrompt-Soft-TF's Parameters** on 10-Split-CIFAR100.

