# OpenReview forum: "Soft-TransFormers for Continual Learning"
_ICLR.cc/2025/Conference — ICLR 2025 Conference Withdrawn Submission_

### Official Review · Reviewer_d7YX · 2024-10-28

**Soundness:** 3
**Presentation:** 3
**Contribution:** 2
**Rating:** 5
**Confidence:** 4

**Summary:**

This paper presents an approach that combines Prompt techniques with learnable soft masks to address catastrophic forgetting in continual learning.  Inspired by Well-initialized Lottery Ticket Hypothesis (WLTH), which provides suboptimal fine-tuning solutions, this paper proposes a fully fine-tuned continual learning (CL) method referred to as Soft-TransFormers (Soft-TF).

**Strengths:**

1. The paper includes extensive experiments across various datasets and tasks, providing comprehensive evidence for the method’s effectiveness.
2. The paper is well-written, with clear organization and logical flow that effectively communicates complex ideas.

**Weaknesses:**

### 1. Insufficient Technical Explanation

**1.1 Lack of Detailed Specification on Soft Mask \( m \) Initialization, Selection, and Optimization**
The paper suggests that the soft mask \( m \) is task-specific, adapting the subnetwork structure based on each task's needs. However, it lacks essential details on how these masks are initialized, selected, and optimized for each task. For instance, are there specific initialization strategies, criteria for mask selection, or task-transition strategies? Such missing details make it challenging to understand the core contribution of the paper, especially as the optimization of these masks is crucial for retaining knowledge in continual learning.

**1.2 Potential Growth in Parameters Due to Task-Specific Masks**
As each task requires an independent soft mask \( m \), the storage cost may grow exponentially with the number of tasks. This is particularly concerning in scenarios with a large number of continual learning tasks, where storage costs could become prohibitive. The paper should discuss how to mitigate this issue.

**1.3 Lack of Clarity on the Setting of \( \alpha_t \)**
The parameter \( \alpha_t \) is referenced but not clearly explained in terms of how it is set or updated for each task. How does the method ensure that the sum of \( \alpha_t \) across tasks equals 1, and what role does this parameter play in mask composition? Detailed explanations and possible constraints for \( \alpha_t \) are necessary for comprehending how the method balances task adaptation and parameter sharing.

**1.4 Redundant Calculation of \( t_x \) in Algorithm 2**
Algorithm 2 calculates \( t_x \) twice, raising the question of whether the indices for the E-Prompt and the learned subnetwork are identical. This redundancy lacks justification, and without clear reasoning, readers may question the necessity of these duplicate operations. A clarification is needed regarding the purpose of this repeated calculation and the roles of E-Prompt and learned subnetworks in relation to \( t_x \).

**1.5 Increased Inference Cost Due to Subnetwork Selection**
According to Algorithm 2, selecting the learned subnetwork for a specific task requires backpropagation for subnetwork selection, which can significantly increase inference costs. This may limit the method's applicability in real-time or resource-constrained environments.

### 2. Unclear Core Mechanism

**2.1 Ambiguity in the Combination Mechanism of Soft Mask \( m \)**
The paper does not clarify why and how soft masks \( m \) can be combined based on different tasks. Ensuring each task's mask is unique and effectively isolated from other tasks is vital to avoid task interference and catastrophic forgetting. The paper lacks an explanation of how such task differentiation is achieved in practice. This is a crucial detail for understanding the validity of the proposed method's mechanism.

**2.2 Ensuring Task-Specific Mask Differentiation**
The authors do not provide details on how task-specific masks are designed to ensure they differ across tasks. If there is overlap or similarity between masks for different tasks, there is a high risk of interference, which could exacerbate forgetting. Further clarification on how each task’s mask maintains uniqueness would strengthen the proposed method.

**Questions:**

I recommend the authors address these concerns by providing detailed descriptions of the soft mask initialization, selection, and optimization processes, as well as methods for managing storage and computational costs with an increasing number of tasks. Clarifying the underlying mechanisms of mask composition and task differentiation would greatly enhance the transparency and reproducibility of the method.

---

> ### Author Response · Authors · 2024-11-17
> **Random Initialized Soft-TF, Parameter-efficeint, and Gradient ID inference**
>
> Thank you for your constructive feedback and concerns. We have carefully prepared the following responses to address all the points raised.
>
> - We have conducted an additional experiment on Random Initialized Performances of Soft-TF to show the importance of well-initialized pre-trained parameters.
>
> - Additionally, we have prepared LLM-based fine-tuning (Adaptor and LoRA) performances to demonstrate the effectiveness of Soft-TF fine-tuning, as shown in **RTables 1, 2, 1-1, 2-1, 3, and 4**.
>
> - Furthermore, we have explained more about Algorithm 2 in detail for clearance of the Soft-TF algorithm.
>
> We believe that the rebuttals have further strengthened our main contribution to the well-initialized parameters of Soft-TF. We hope we have addressed the reviewer's major concerns, and we are happy to answer any further questions or concerns.
>
> Best regards,
>
> Authors

---

> ### Author Response · Authors · 2024-11-17
> **(RTable. 5) Random initalized Performances of Soft-TF**
>
> |             |              |           |             | **10-Split-CIFAR100** ||
> | ----------- | :---------:  | :-------: | :-----:     | :------: | :-----: |
> | **Method**                             | Pretrained-Dataset | Task ID   | Random Init.| ACC (&uarr;) |  Forget (&darr;) |
> | DualPrompt-Soft-TF-L[10,11,12] w/o FF  | ImageNet-21K | Prompt ID | Xaiver | 90.59 | 3.85 |
> | DualPrompt-Soft-TF-L[10,11,12] w/o FF  | ImageNet-21K | Prompt ID | Kaiming | 90.72 | 3.63 |
> | DualPrompt-Soft-TF-L[10,11,12] w/o FF  | ImageNet-21K | Prompt ID | Normal | 90.45 | 3.78 |
> | DualPrompt-Soft-TF-L[10,11,12] w/o FF  | ImageNet-21K | Prompt ID | **Uniform(1.0, 1.0)** | 92.35 | 2.98 |
> | DualPrompt-Soft-TF-L[10,11,12] w/o FF  | ImageNet-21K | **Gradient ID** | **Uniform(1.0, 1.0)** | **98.05** | **0.25** |

---

> ### Author Response · Authors · 2024-11-17
> **Random Initalized Performances of Soft-TF**
>
> - Random initialization of Soft-Transformer's weights plays a critical role when leveraging well-pretrained models like Vision Transformers (ViTs).
> - The optimal training point is the parameters of a well-pretrained model. Among the initialization methods, Uniform initialization for Soft-TransFormer satisfies this requirement effectively.
> - To validate our claims, we analyze the impact of common random initialization methods, including Xavier, Kaiming, Normal, and Uniform Initialization, as shown in RTable. 5.
> - The results demonstrate that the same well-initialization point leads to independent optimal task performance, particularly with Gradient ID inference.
> - Furthermore, this ablation study has strengthened our Soft-TF with state-of-the-art performances inspired by the Well-initialized Lottery Ticket Hypothesis (WLTH), showing better performances than LLMs (Adapter & LoRA) in RTable. 4.

---

> ### Author Response · Authors · 2024-11-17
> **Finetuning Parameters of Adapter, LoRA, and Soft-TF**
>
> - As **RTable. 4** (our rebuttal) stated, the LLM fine-tuning method such as Adapter and LoRA (6.91M) requires additional parameters in the pre-trained models to optimize end-tasks, including language and vision tasks.
> - In our optimization process, as shown in **RTable. 2**, only the final 12 attention layers (Soft-TF-L[12] with 2.3M) were tuned to obtain performance close to the best in accuracy and training times. The trainable parameter 2.3M per task corresponds to 3% of the pre-trained model parameters 68.5M, which is negligible.
> - Furthermore, the parameter-efficient fine-tuning of Soft-TF was experimentally proven to achieve equivalent or par performance with only WSN, c=90% parameters. This will significantly contribute to parameter-efficient CL models.

---

> ### Author Response · Authors · 2024-11-17
> **Gradient ID Inference in Alg. 2**
>
> ### On $\alpha_t$ in Algorithm 2
> - The value of $\alpha_t$ is determined by 1/the number of all seen tasks $T$, ensuring that $\alpha_t$ assigns equal importance to all seen tasks.
> - Subsequently, by applying the entropy loss and computing gradient w.r.t. $\alpha_t$, the most sensitive task_id ($t_x$) can be obtained.
>
>
> ### Redundant Calculation of ($t_x$)
>
> - In Algorithm 2, we select one of the task inference methods: (1) Prompt ID or (2) Gradient ID.
>
> - During the inference step, if Gradient ID is selected, we use $t_x$ inferred by the Gradient ID algorithm to identify the corresponding e-prompt $e_{t_x}$ and soft-TF $m_{t_x}$.
> ​
> - Extensive experiments prove that $e_{t_x}$ is a minor factor in determining Gradient ID, as trained e-prompts are not perfect solutions for specific tasks. Instead, $m_{t_x}$, the primary tuning parameter, plays a crucial role. All empirical results consistently support this conclusion, i.e., the contribution of task inference:  $m_{t_x} \gg e_{t_x}$.
>
> - In the Prompt ID method, we use $t_x$ estimated by Prompt ID as the index to retrieve both the e-prompt $e_{t_x}$ and soft-TF $m_{t_x}$.
>
> ### Few-shot Gradient ID Inference
> - The 3-shot Gradient ID inference of Soft-TF is the most efficient approach regarding performance and inference speed, as shown in **RTable. 2-1**. Please refer to **RTables 1-1** and **2-1** for more information on the time complexity of few-shot Gradient ID inference.

---

> ### Author Response · Authors · 2024-11-17
> **Clarity on Soft-TF**
>
> - As outlined in the Uniform Initialized performances (RTable 5), Soft-TF independently learns each task under identical initialization conditions (using non-overlapped datasets) with task IDs provided during CIL training.
>
> - By beginning with the same initialization point, Soft-TF can discover a unique solution $m_t$ for each task by learning from its corresponding independent category dataset.
>
> - This capability of Soft-TF enables Gradient ID Inference to overcome the limitations of Prompt ID effectively.
>
> - Extensive experiments on continual learning scenarios involving repeated or similar category datasets are beyond the scope of this study and will be addressed in future work. According to our expectations, Soft-TF's performance on overlapped dataset continual scenarios could range from [Prompt ID] to [Gradient ID with 100% inference accuracy].

---

> > ### Comment · Reviewer_d7YX · 2024-12-03
> >
> > From your reply, it seems that Soft-TF requires setting a set of task-specific masks for each task, which raises the question: How does this differ from traditional model architecture extension methods in incremental learning? As the number of tasks increases, the number of parameters required for prediction is increasing?

---

> ### Author Response · Authors · 2024-12-03
>
> We acknowledge reviewer d7YX’s observation that architecture-based continual learners increase model parameters as the number of tasks grows.
>
> However, similar to Soft-TF, large language models (LLMs) such as LoRA and Adapter also increase the parameters when fine-tuning sequential tasks, as shown in **RTable 4**. Notably, Soft-TF outperforms all existing baselines (LoRA and Adapter) by **a margin of 6%** under the same number of tunable parameters (6.9M), demonstrating superior performance.
>
> Additionally, as suggested by reviewer ygEK, we conducted an ablation study to further highlight the novelty of Soft-TF, with results presented in **RTable 3** and **4** (Comparisons of Soft-TF with LLMs). Soft-TF exhibits two key distinctive features compared to conventional baselines like LoRA and Adapter:
>
> (1) **Direct-full fine-tuning at the sparse attention layer:** Soft-TF is well-initialized at the last three layers or **one single layer (see RTable. 6)** and fine-tuned with pre-trained parameters directly for all end tasks. In contrast, LoRA fine-tunes a small set of low-rank parameters across all attention layers, while the Adapter introduces additional modules in the FeedForward (FF) Network layer to enable representation learning for sequential tasks.
>
> (2) **Efficiency in Fine-tuning:** Our extensive experiments demonstrate that fine-tuning Soft-TF on three attention layers is sufficient for sequential tasks. By comparison, LoRA and Adapter require fine-tuning across all layers. From our observations, LoRA and Adapter failed to fully fine-tune all sequential tasks, as shown in **RTable. 6**.
>
> In conclusion, **as an innovative novel method (the first approach in CL)**, we have demonstrated the effectiveness of Soft-TF with sparse full fine-tuning, surpassing conventional fine-tuning methods, including LoRA and Adapter. Our work has introduced a novel full fine-tuning approach that preserves the integrity of the pre-trained model's parameters while achieving superior performance.
>
> Sincerely,
>
> The authors.

---

> > ### Author Response · Authors · 2024-12-03
> >
> > Dear reviewer d7YX,
> >
> > We hope our responses have thoroughly addressed the reviewer’s questions and concerns.
> >
> > We would clarify or provide further details if you have additional feedback or questions.
> >
> > If you think that all concerns have been adequately resolved, please consider updating your official score correspondingly.
> >
> > Thank you for your time and effort in reviewing our work.
> >
> > Sincerely,
> >
> > The Authors

---

> ### Author Response · Authors · 2024-12-03
> **(RTable. 6) Trainable Parameter-wise Comparison of Soft-TF with LLMs (Adapter & LoRA) in terms of Params, ACC and Forget**
>
> |             | **ViT-B/12 (85.8M)** |   |    **10-Split-CIFAR100**    || **20-Split-CIFAR100**      || **10-Split-ImageNet-R**   ||
> | ----------- | :---------: | :-------: | :---------:    | :---------: | :-----------: | :---------: | :----------: | :----------: |
> | **Method**  | Train Params | Task ID   | ACC (&uarr;) |  Forget (&darr;) | ACC (&uarr;) |  Forget (&darr;) | ACC (&uarr;) |  Forget(&darr;) |
> | Adapter-L[10,11,12] w/ FF, r=1 | 0.09M | Prompt ID | 86.38 | 4.87 | 85.61 | 5.04 | 70.95 | 4.31 |
> | Adapter-L[10,11,12] w/ FF, r=4 | 0.36M | Prompt ID | 86.53 | 4.52 | 85.66 | 5.00 | 70.82 | 4.90 |
> | **Adapter**-L[10,11,12] w/ FF, r=75 | **6.91M** | Prompt ID | 86.45 | 4.61 | 84.75 | 5.11 | 70.55 | 4.74 |
> | LoRA-L[10,11,12] w/o FF, r=4 | 0.06M | Prompt ID | 82.19 | 4.33 | 93.74 | 2.07 | 70.90 | 9.11 |
> | LoRA-L[10,11,12] w/o FF, r=24 | 0.32M | Prompt ID | 86.77 | 4.27 | 95.65 | 1.04 | 69.81 | 10.30 |
> | **LoRA**-L[10,11,12] w/o FF, r=500 | **6.91M** | Prompt ID | 82.00 | 4.33 | 92.14 | 2.02 | 43.51 | 13.21 |
> | **Soft-TF**-L[10,11,12] w/o FF | **6.93M** | Prompt ID |  **92.35** | **2.98** | **97.40** | **0.57**| **76.62** | **5.30** |
> | **Soft-TF**-L[12] w/o FF | 2.31M | Gradient ID |  **96.84** | **0.55** | **97.81** | **0.57** | **81.18** | **1.31** |

---

### Official Review · Reviewer_ygEK · 2024-10-31

**Soundness:** 2
**Presentation:** 3
**Contribution:** 2
**Rating:** 5
**Confidence:** 3

**Summary:**

This work proposed sub-network over the prompt tuning technique for continuous learning.
The overall presentation is clear but the motivation and the novelty is unclear.
Experimental results demonstrate superior performance to other methods.

**Strengths:**

The overall presentation is clear.
The experimental results demonstrate the effectiveness of the proposed method.

**Weaknesses:**

Firstly, the motivation is unclear.  In lines 84-85, the paper stated that " It cannot capture all the nuances of uncorrelated sequential tasks wildly if the task significantly differs from what the CL model was pre-trained initially on". It seems this work aims to address the reliance on a well pretraining, but the proposed method also require a well-initialized network as stated in lines 96-99.
Besides, in lines 224-225, the task-specific prompts can be regarded as the explicit task-specific fine-tuning.

Secondly, the proposed method seems to be a combination of parameter-efficient fine-tuning method, e.g., prompt + LoRA, prompt + Adapter. How much performance improvement would be caused by adding LoRA or adapters for specific tasks in the baseline method。
The superior performance seems contributed by more learnable parameters.

The experimental results requires more analysis.
(i) Explain the ``Upper bound''. In fact, since the proposed method modifies the model architecture, the ``Upper bound''  cannot be the same as the other paper.
(ii) Why the proposed method achieves higher performance than the ``Upper bound''?
(iii) Why the proposed method achieves higher performance when using DualPrompt as baseline while the improvement over the L2P seems smaller.
(iv). In table 1, why the performance under the  20-Split-CIFAR100 is higher than  10-Split-CIFAR100? In general, the more incremental learning step, the weaker the performance due to forgetting. I suspect that the improved performance is due to learning more parameters. The number of parameters should be reported.

**Questions:**

Please refer to the weakness.

---

> ### Author Response · Authors · 2024-11-17
> **Comparisions Soft-TF with LLMs (Adapter & LoRA)**
>
> Thank you for your constructive feedback and concerns. We have carefully prepared the following responses to address all the points raised as follows:
>
> - We have compared Soft-TF with recommended LLMs (Adapter and LoRA) with the same trainable parameters (**RTables 3 and 4**), showing outperformance (the most parameter-efficient architecture, Soft-TF) in terms of accuracy and forgetting.
>
> - We have added the upper bound of Soft-TF for fair comparisons.
>
> - We explain the reasons for the performances of DualPrompt-Soft-TF v.s. L2P-Soft-TF.
>
> - We have revised the Motivation of Soft-TF as the reviewer suggested.
>
> We believe the rebuttals have strengthened our main contribution through extensive investigations. We hope we have addressed the reviewer's major concerns, and we are happy to answer any further questions or concerns.
>
> Best regards,
>
> Authors

---

> ### Author Response · Authors · 2024-11-17
> **(RTable. 3) Comparisions of Soft-TF with LLMs (Adapter & LoRA) Soft-TF parameters, training and test time**
>
> |             | **ViT-B/12 (85.8M)** |   |    **10-Split-CIFAR100**    || **20-Split-CIFAR100**      || **10-Split-ImageNet-R**   ||
> | ----------- | :---------: | :-------: | :---------:    | :---------: | :-----------: | :---------: | :----------: | :----------: |
> | **Method**  | Train Params | Task ID   | Train(sec.) |  Test(sec.) | Train(sec.) |  Test(sec.) | Train(sec.) |  Test(sec.) |
> | DualPrompt  | 0.00M        | Prompt ID | 12.12K       | 76           | 11.60K       | 78           | 13.10K       | 47          |
> | &nbsp; PGP  | 0.00M        | Prompt ID | 12.21K       | 76           | 13.12K       | 78           | 13.33K       | 47           |
> | &nbsp; **Adapter**-L[10,11,12] w/ FF, r=75 | 6.91M | Prompt ID | 13.66K | 88 | 15.72K | 106 | 15.50K | 53 |
> | &nbsp; **LoRA**-L[10,11,12] w/o FF, r=500 | 6.91M | Prompt ID | 13.24K | 79  | 15.89K  | 105  | 15.09K | 53 |
> | &nbsp; **Soft-TF**-L[10,11,12] w/o FF | 6.93M | Prompt ID   | 13.87K | 80  | 15.60K | 104 | 15.35K | 52 |

---

> ### Author Response · Authors · 2024-11-17
> **(RTable. 4) Comparisions of Soft-TF with LLMs (Adapter & LoRA) in terms of Params, ACC and Forget**
>
> |             | **ViT-B/12 (85.8M)** |   |    **10-Split-CIFAR100**    || **20-Split-CIFAR100**      || **10-Split-ImageNet-R**   ||
> | ----------- | :---------: | :-------: | :---------:    | :---------: | :-----------: | :---------: | :----------: | :----------: |
> | **Method**  | Train Params | Task ID   | ACC (&uarr;) |  Forget (&darr;) | ACC (&uarr;) |  Forget (&darr;) | ACC (&uarr;) |  Forget (&darr;) |
> | DualPrompt | 0.00M | Prompt ID | 86.50 | 5.77 | 82.98 | 8.20  | 68.13  | 4.46   |
> | &nbsp; PGP | 0.00M | Prompt ID | 86.92 | 5.35 | 83.74 | 7.91  |  69.34 |  4.53   |
> | &nbsp; **Adapter**-L[10,11,12] w/ FF, r=75 | 6.91M | Prompt ID | 86.45 | 4.61 | 84.75 | 5.11 | 70.55 | 4.74 |
> | &nbsp; **LoRA**-L[10,11,12] w/o FF, r=500 | 6.91M | Prompt ID | 82.00 | 4.33 | 92.14 | 2.02 | 43.51 | 13.21 |
> | &nbsp; **Soft-TF**-L[10,11,12] w/o FF | 6.93M | Prompt ID |  **92.35** | **2.98** | **97.40** | **0.57** | **76.62** | **5.30** |
> | Upper-bound of Soft-TF | 6.93M | - |  93.90 | - | 93.90 | - | 80.21 | - |

---

> ### Author Response · Authors · 2024-11-17
> **Comparisions of Soft-TF with LLMs (Adapter and LoRA)**
>
> - To demonstrate the effectiveness of Soft-TF, we compare Soft-TF against LLM fine-tuning methods such as Adapters and LoRA, as shown in **RTables 3 and 4**.
>
> - Under identical experimental conditions — including trainable model parameters (~6.9M per task), layers (L[10,11,12]), and Prompt ID — Soft-TF outperformed other LLM-based fine-tuning approaches, as shown in **Rtable. 4**. The results highlight that directly updating well-pretrained model parameters and prompt-tuning via Soft-TF is more effective than combining representations through LoRA or learning representations with Adapters.
>
> - Furthermore, we observed that Soft-TF and the other methods exhibited comparable training and testing time complexity for a similar number of trainable parameters. Notably, single-layer fine-tuning using Soft-TF (with L[12]) surpasses the baselines' performances.
>
> - These findings firmly establish Soft-TF as the most competitive approach among strong LLM baselines (Adapters and LoRA) in the continual learning (CIL) scenario.
>
> - We have reported more parameter-wise performances of Adapter and LoRA in the updated script.

---

> ### Author Response · Authors · 2024-11-17
> **Upper-bound of Soft-TF & Performances on 10/20-Split-CIFAR100**
>
> - As demonstrated above, in **RTable. 4**, Soft-TF outperforms baseline approaches such as LLM-based methods (Adapter and LoRA) by providing more optimal solutions at attention layers for specific sub-tasks.
>
> - Soft-TF can exceed the upper-bound performance because joint learning (upper-bound) requires simultaneous training across the entire category space with fixed parameters (e.g., 100-way classification in CIFAR-100). In contrast, under a Class-Incremental Learning (CIL) scenario, Soft-TF focuses on task-specific training (e.g., 10-way classification in 10-Split-CIFAR-100 or 5-way classification in 20-Split-CIFAR-100). This focused approach allows Soft-TF to handle 5/10-way classification tasks more effectively than 100-way ones (joint training). The 5/10-way classification problem can also be interpreted in this context. From a Soft-TF perspective, the 5-way classification problem is more accessible than the 10-way classification problem.

---

> ### Author Response · Authors · 2024-11-17
> **Perfomances of DualPrompt-Soft-TF v.s. L2P-Soft-TF**
>
> - Clarifying the structural differences between DualPrompt and L2P is essential to interpreting the performance differences between DualPrompt-L2P and DualPrompt-Soft-TF.
>
> - L2P employs a straightforward structure where prompt embeddings are directly fed into a pre-trained ViT model. In contrast, DualPrompt incorporates global prompt tuning (G-Prompt) in the initial layers (layers 1 and 2) to adapt to task-specific domains. Additionally, DualPrompt applies task-specific prompt tuning (E-Prompt) in specific attention layers to learn specialized sub-tasks. Prior studies have demonstrated that the combined G/E-Prompt tuning strategy in DualPrompt consistently outperforms the simple L2P approach.
>
> - This study of Soft-TF builds upon DualPrompt's training philosophy, utilizing G/E-Prompt as baselines. However, only DualPrompt fails to achieve optimal sequential task learning in continual learning (CL) scenarios. To address these limitations, we propose Soft-TF tuning, which enhances performance by updating pre-trained model parameters through soft-learnable parameters alongside E-Prompt tuning. This approach achieves superior results since E-Prompt and Soft-TF provide more task-specific optimal solutions.

---

> ### Author Response · Authors · 2024-11-17
> **Revised Motivation of Soft-TF**
>
> - To clarify the motivation of Soft-TF, we have revised the sentences as follows:
>
> **"The only prompt-tuning of the pre-trained model cannot capture all the nuances of uncorrelated sequential tasks even though leveraging the frozen well-initialized model pre-trained on large-scale datasets since the only frozen well-initialized model provides global solutions rather than task-specific solutions provided by fine-tuning."**
>
> - We have revised the motivation in our updated script.

---

### Official Review · Reviewer_Rgmo · 2024-11-04

**Soundness:** 3
**Presentation:** 3
**Contribution:** 3
**Rating:** 5
**Confidence:** 4

**Summary:**

In this paper, the authors attempt to solve the continual learning problem via a soft transformer idea, which adaptively selects subnetworks (specified by learnable masking for pre-trained parameters) and soft network tasked for detailed tasks in CL. During the sequential learning, the model will be trained to achieve a task-specific soft network based on the task-agnostic pre-trained model and sparsely activate some weights in the layers or deactivate someone. Besides, the authors support some theory proofs based on Well-initialized Lottery Ticket Hypothesis to analyze the rationale behind the improvements. Implemented on baselines, like DualPrompt, L2P, and L2p-PGP, they obtain competitive even SoTA performances.

**Strengths:**

1. The presentation sounds well.

2. The final experimental results present competitive even SoTA on some evaluated tasks.

3. Implementing the method on the Prompt Pool, which can divide the prompt pool into several groups and enable different group prompts to acquire different knowledge sounds reasonable while applying soft-network or binary masks on pre-trained weights can further boost the training.

**Weaknesses:**

1. Somehow, I believe such soft-network or subnetwork ideas have been widely studied among the communities, like model compression, model purification, or PEFT. One can think of employing the extra prompt vectors or using a learnable masking strategy, forcing the model to fit the downstream task while the pre-trained weights are fixed to hold the generalizability. Moreover, the prompt pool construction mechanism has been widely studied by DualPrompt, L2P.

2. I am curious about the training efficiency and inference latency since they utilize a large subnetwork or softnework compared with the baseline.

3. Missing the tunable parameter comparisons with baselines and existing methods.

**Questions:**

Please refer to my above weaknesses.

---

> ### Author Response · Authors · 2024-11-17
> **(RTable. 1) Soft-TF parameters, training and test time without FeedForward (FF) networks.**
>
> Thank you for your constructive feedback and concerns. To address all the points raised, we have prepared the following responses. We have clarified the proposed **parameter-efficient Soft-TF** through extensive experiments and further emphasized its novel contributions as follows:
>
> - We have investigated layer-wise training efficiency & inference of Soft-TF (**RTables 1 and 2**) with gradient and prompt ID: only prompt ID-based Soft-TF outperformed strong baselines such as PGP, Adapter, and LoRA.
>
> - Furthermore, we have observed that the 3-shot Gradient ID inference cost (using samples within a mini-batch) with the last layer (Soft-TF-L[12]) is approximately 1.1 times that of Prompt ID, leading to the most efficient Soft-TF structure, as shown in **RTables 1-1 and 2-1**.
>
> We hope we have addressed the reviewer's major concerns, and we are happy to answer any further questions or concerns.
>
> Best regards,
>
> Authors
>
> |             | **ViT-B/12 (85.8M)** |   |    **10-Split-CIFAR100**    || **20-Split-CIFAR100**      || **10-Split-ImageNet-R**   ||
> | ----------- | :---------: | :-------: | :---------:    | :---------: | :-----------: | :---------: | :----------: | :----------: |
> | **Method**  | Train Params | Task ID   | Train (sec.) |  Test (sec.) | Train (sec.) |  Test (sec.) | Train (sec.) |  Test (sec.) |
> | DualPrompt  | 0.00M        | *Prompt ID* | *12.12K*   | *76*  | *11.60K*       | *78*  | *13.10K* | 47          |
> | &nbsp; PGP  | 0.00M        | Prompt ID | 12.21K       | 76           | 13.12K       | 78        | 13.33K   | 47           |
> | &nbsp; Soft-TF-L[12] w/o FF       | 2.31M | Gradient ID | 12.24K | 103 | 13.40K | 132 | 13.42K | 66 |
> | &nbsp; Soft-TF-L[11,12] w/o FF    | 4.62M | Gradient ID | 12.95K | 115 | 14.38K | 146 | 14.23K | 73 |
> | &nbsp; **Soft-TF**-L[10,11,12] w/o FF | 6.93M | **Gradient ID** | **13.71K** | **130** | **15.51K** | **163** | **15.08K** | **82** |
> | &nbsp; **Soft-TF**-L[10,11,12] w/o FF | 6.93M | *Prompt ID*   | *13.87K* | *80*  | *15.60K* | *104* | *15.35K* | *52* |
> | &nbsp; Adapter-L[10,11,12] w/ FF, r=75 | 6.91M | Prompt ID | 13.66K | 88 | 15.72K | 106 | 15.50K | 53 |
> | &nbsp; LoRA-L[10,11,12] w/o FF, r=500 | 6.91M | Prompt ID | 13.24K | 79  | 15.89K  | 105  | 15.09K | 53 |

---

> ### Author Response · Authors · 2024-11-17
> **Training Efficiency & Inference**
>
> - We use DualPrompt and DualPrompt-PGP as baselines to evaluate the parameter efficiency of Soft-TF.
> - As the number of Soft-TF layers increases, as shown in **RTable. 1**, the number of trainable parameters increases slightly, and task performance improves correspondingly.
>
> - Gradient ID's inference (mini-batch) complexity is reported to be approximately 1.6 times that of Prompt ID; however, Gradient ID boosts performance and reduces forgetting dramatically.
>
> - The Prompt ID inference complexity of Soft-TF-L[10,11,12] with three layers is comparable to the baselines (PGP) while delivering superior performance.
>
> - Notably, even single-layer fine-tuning using Soft-TF (with L[12], 2.3M) outperformed the baselines (PGP) with comparable training time complex, as shown in **RTable. 2**.
>
> - Moreover, as outlined in our script, sparse learning techniques such as WSN (with c=90%) can be applied to Soft-TF-L[12], significantly reducing the number of tunable parameters (2.0M) while maintaining Soft-TF's high performances.
>
> - To show Soft-TF's effectiveness further, we have compared it with LLM fine-tuning methods such as Adapter and LoRA, as demonstrated in RTables 3 and 4. We have included all ablation studies in our updated script.
>
>
> - **(Time Complexity of Gradient ID )**: We investigate the most parameter-efficient and gradient-based task inference methods, as shown in **RTables 1-1 and 2-1**. - Our findings reveal that the 3-shot Gradient ID inference cost (using samples within a mini-batch) with the last layer (**Soft-TF-L[12], 2.3M**) is approximately **1.1 times that of Prompt ID**, maintaining comparable efficiency while delivering **superior performance**. Note that m-batch denotes mini-batch.

---

> ### Author Response · Authors · 2024-11-17
> **(RTable. 2) Soft-TF parameters, Accuracy, and Forget without FeedForward (FF) networks.**
>
> |             | **ViT-B/12 (85.8M)** |   |    **10-Split-CIFAR100**    || **20-Split-CIFAR100**      || **10-Split-ImageNet-R**   ||
> | ----------- | :---------: | :-------: | :---------:    | :---------: | :-----------: | :---------: | :----------: | :----------: |
> | **Method**  | Train Params | Task ID   | ACC (&uarr;) |  Forget (&darr;) | ACC (&uarr;) |  Forget (&darr;) | ACC (&uarr;) |  Forget (&darr;) |
> | DualPrompt | 0.00M | *Prompt ID* | *86.50* | *5.77* | *82.98* | *8.20*  | *68.13*  | *4.46*   |
> | &nbsp; PGP | 0.00M | Prompt ID | 86.92 | 5.35 | 83.74 | 7.91  |  69.34 |  4.53   |
> | &nbsp; Soft-TF-L[12] w/o FF | 2.31M  | Gradient ID | 96.84 | 0.55 | 97.81 | 0.57 | 81.18 | 1.31 |
> | &nbsp; Soft-TF-L[11,12] w/o FF | 4.62M | Gradient ID | 97.58 | 0.34 | 98.65 | 0.43 | 83.09 | 0.42 |
> | &nbsp; **Soft-TF**-L[10,11,12] w/o FF | 6.93M | **Gradient ID** | **98.05** | **0.25** | **98.96** | **0.23** | **83.70** | **0.53** |
> | &nbsp; **Soft-TF**-L[10,11,12] w/o FF | 6.93M | *Prompt ID* |  *92.35* | *2.98* | *97.40* | *0.57* | *76.62* | *5.30* |
> | &nbsp; Adapter-L[10,11,12] w/ FF, r=75 | 6.91M | Prompt ID | 86.45 | 4.61 | 84.75 | 5.11 | 70.55 | 4.74 |
> | &nbsp; LoRA-L[10,11,12] w/o FF, r=500 | 6.91M | Prompt ID | 82.00 | 4.33 | 92.14 | 2.02 | 43.51 | 13.21 |

---

> ### Author Response · Authors · 2024-11-20
> **(RTable. 1-1) Soft-TF (Gradient ID) parameters, training and test time without FeedForward (FF)" networks.**
>
> |             | **ViT-B/12 (85.8M)** |   |    **10-Split-CIFAR100**    || **20-Split-CIFAR100**      || **10-Split-ImageNet-R**   ||
> | ----------- | :---------: | :-------: | :---------:    | :---------: | :-----------: | :---------: | :----------: | :----------: |
> | **Method**  | Train Params | Task ID   | Train (sec.) |  Test (sec.) | Train (sec.) |  Test (sec.) | Train (sec.) |  Test (sec.) |
> | DualPrompt  | 0.00M        | *Prompt ID* | *12.12K*   | *76*  | *11.60K*       | *78*  | *13.10K* | 47          |
> | &nbsp; PGP  | 0.00M        | Prompt ID | 12.21K       | 76           | 13.12K       | 78        | 13.33K   | 47           |
> | &nbsp; Soft-TF-L[12] w/o FF       | 2.31M | Prompt ID | 12.24K | 79 | 13.40K | 80 | 13.42K | 48 |
> | &nbsp; Soft-TF-L[12] w/o FF       | 2.31M | Gradient ID, **3-shot** | 12.24K | **88** | 13.40K | **90** | 13.42K | **57** |
> | &nbsp; Soft-TF-L[12] w/o FF       | 2.31M | Gradient ID, 5-shot | 12.24K | 94 | 13.40K | 98 | 13.42K | 61 |
> | &nbsp; Soft-TF-L[12] w/o FF       | 2.31M | Gradient ID, 7-shot | 12.24K | 95 | 13.40K | 108 | 13.42K | 62 |
> | &nbsp; Soft-TF-L[12] w/o FF       | 2.31M | Gradient ID, m-batch | 12.24K | 103 | 13.40K | 132 | 13.42K | 66 |

---

> ### Author Response · Authors · 2024-11-20
> **(RTable. 2-1) Soft-TF (Gradient ID) parameters, Accuracy, and Forget without FeedForward (FF) networks.**
>
> |             | **ViT-B/12 (85.8M)** |   |    **10-Split-CIFAR100**    || **20-Split-CIFAR100**      || **10-Split-ImageNet-R**   ||
> | ----------- | :---------: | :-------: | :---------:    | :---------: | :-----------: | :---------: | :----------: | :----------: |
> | **Method**  | Train Params | Task ID   | ACC (&uarr;) |  Forget (&darr;) | ACC (&uarr;) |  Forget (&darr;) | ACC (&uarr;) |  Forget (&darr;) |
> | DualPrompt | 0.00M | *Prompt ID* | *86.50* | *5.77* | *82.98* | *8.20*  | *68.13*  | *4.46*   |
> | &nbsp; PGP | 0.00M | Prompt ID | 86.92 | 5.35 | 83.74 | 7.91  |  69.34 |  4.53   |
> | &nbsp; Soft-TF-L[12] w/o FF | 2.31M  | Prompt ID | 91.83 | 2.99 | 96.43 | 1.00 | 72.45 | 5.32 |
> | &nbsp; Soft-TF-L[12] w/o FF | 2.31M  | Gradient ID, **3-shot** | **93.12** | **1.82** | **96.43** | **1.00** | **73.55** | **4.80** |
> | &nbsp; Soft-TF-L[12] w/o FF | 2.31M  | Gradient ID, 5-shot | 96.13 | 0.58 | 96.43 | 1.00 | 75.04 | 4.49 |
> | &nbsp; Soft-TF-L[12] w/o FF | 2.31M  | Gradient ID, 7-shot | 96.51 | 0.65 | 96.43 | 1.00 | 76.34 | 4.75 |
> | &nbsp; Soft-TF-L[12] w/o FF | 2.31M  | Gradient ID, m-batch | 96.84 | 0.55 | 97.81 | 0.57 | 81.18 | 1.31 |

---

### Author Response · Authors · 2024-11-21
**Our responses to the concerning points**

# Dear All Reviewers and Chairs

Thank you for your constructive feedback and concerns. We have prepared the following responses to address all the points raised by reviewers. We have clarified the proposed **parameter-efficient Soft-TF** through extensive experiments and further emphasized its novel contributions as follows:

### For **Reviewer Rgmo**,

- We have investigated layer-wise training efficiency & inference of Soft-TF (**RTables 1 and 2**) with gradient and prompt ID: only prompt ID-based Soft-TF outperformed strong baselines such as PGP, **Adapter**, and **LoRA**.

- Furthermore, we have observed that the 3-shot Gradient ID inference cost (using samples within a mini-batch) with the last layer (Soft-TF-L[12]) is approximately 1.1 times that of Prompt ID, leading to the most efficient Soft-TF structure, as shown in **RTables 1-1 and 2-1**.

### For **Reviewer ygEK**,
- We have compared Soft-TF with recommended LLMs (**Adapter** and **LoRA**) with the same trainable parameters (**RTables 3 and 4**), showing outperformance (the most parameter-efficient architecture, Soft-TF) regarding accuracy and forgetting.

- We have added the upper bound of Soft-TF for fair comparisons.

- We have explained **why DualPrompt-Soft-TF outperforms  L2P-Soft-TF** with backgrounds of DualPrompt of Global Prompt and Local Prompts (E-Prompts).

- We have revised the Motivation of Soft-TF as the reviewer suggested.

### For **Reviewer d7YX**,

- We have conducted an additional experiment on Random Initialized Performances of Soft-TF to show the importance of well-initialized pre-trained parameters (**see RTable. 5**).

- Particularly in the **Author-Reviewer discussion period**, we have further clarified the difference between our Soft-TF and conventional Transformer fine-tuning baselines such as **Adapter** and **LoRA**, suggesting parameter-wise superior performances (SOTA, **see RTable. 6**) in Accuracy, Forgetting, training, and time efficiency under the comparable trainable parameter settings.

- Furthermore, we have explained more about Algorithm 2 in detail to clarify the Soft-TF algorithm.

Throughout these rebuttals, we have further strengthened our main contribution to **the well-initialized parameter-efficient Soft-TF, outperforming the representative LLM-finetuning methods** (**Adapter** and **LoRA**).

In summary, to our knowledge, **a well (uniformly)-initialized Soft-TF** (mask network) represents **the first approach** to pre-training transformer-based full fine-tuning methods within the Continuous Learning (CL) domain, including applications involving LLM-based techniques.

Best regards,

The Authors

---

> ### Author Response · Authors · 2024-11-27
>
> Dear Reviewers,
>
> We would like to express our heartfelt gratitude for your dedicated time and effort in reviewing our paper. Your constructive feedback has been instrumental in enhancing the quality and clarity of our work.
>
> If you have any further concerns regarding our response, please do not hesitate to share them. We would be delighted to engage in follow-up discussions and address any additional comments you may have.
>
> Sincerely,
> The Authors

---

> ### Author Response · Authors · 2024-12-03
>
> Dear reviewers,
>
> We hope our responses have thoroughly addressed the reviewer’s questions and concerns.
>
> We would clarify or provide further details if you have additional feedback or questions.
>
> If you think that all concerns have been adequately resolved, please make sure to update your official score correspondingly.
>
> Thank you for your time and effort in reviewing our work.
>
> Sincerely,
>
> The Authors

---

### Note · Authors · 2025-01-22

I have read and agree with the venue's withdrawal policy on behalf of myself and my co-authors.